# Spectral State Space Models

## Abstract

This paper studies sequence modeling for prediction tasks with long range dependencies. We propose a new formulation for state space models (SSMs) based on learning linear dynamical systems with the spectral filtering algorithm (Hazan et al., 2017). This gives rise to a novel sequence prediction architecture we call a spectral state space model.

Spectral state space models have two primary advantages. First, they have provable robustness properties as their performance depends on neither the spectrum of the underlying dynamics nor the dimensionality of the problem. Second, these models are constructed with fixed convolutional filters that do not require learning while still outperforming SSMs in both theory and practice.

The resulting models are evaluated on synthetic dynamical systems and long-range prediction tasks of various modalities. These evaluations support the theoretical benefits of spectral filtering for tasks requiring very long range memory.

## 1 Introduction

Handling long-range dependencies efficiently remains a core problem in sequence prediction/modelling. Recurrent Neural Networks (RNN) (Hopfield, 1982; Rumelhart et al., 1985; Elman, 1990) are a natural choice, but are notoriously hard to train; they often suffer from vanishing and exploding gradients (Bengio et al., 1994; Pascanu et al., 2013) and despite techniques to mitigate the issue (Hochreiter & Schmidhuber, 1997; Cho et al., 2014; Arjovsky et al., 2016), they are also hard to scale given the inherently sequential nature of their computation.

In recent years, transformer models (Vaswani et al., 2017) have become the staple of sequence modelling, achieving remarkable success across multiple domains (Brown et al., 2020; Dosovitskiy et al., 2020; Jumper et al., 2021). Transformer models are naturally parallelizable and hence scale significantly better than RNNs. However, attention layers have memory/computation requirements that scale quadratically with context length. Many approximations have been proposed (see Tay et al. (2022) for a recent survey).

RNNs have seen a recent resurgence in the form of state space models (SSM) which have shown promise in modelling long sequences across varied modalities (Gu et al., 2021a; Dao et al., 2022; Gupta et al., 2022; Orvieto et al., 2023; Poli et al., 2023; Gu & Dao, 2023). SSMs use linear dynamical systems (LDS) to model the sequence-to sequence transform by evolving the internal state of a dynamical system according to the dynamics equations

$$x_t = Ax_{t-1} + Bu_t \qquad y_t = Cx_t + Du_t.$$

Here $x_t \in \mathbb{R}^d$ is the hidden state of the dynamical system, $u_t$ is the input to the system, and $y_t$ are observations. The matrices $A, B, C, D$ govern the evolution of the system and are called system matrices. Despite its simplicity, this linear model can capture a rich set of natural dynamical systems in engineering and the physical sciences due to the potentially large number of hidden dimensions. Linear dynamical systems are also attractive as a sequence model because their structure is amenable to both fast inference and fast training via parallel scans (Blelloch, 1989; Smith et al., 2023) or convolutions (Gu et al., 2021a). A rich literature stemming from control theory and recent machine learning interest has given rise to efficient techniques for system identification, filtering, and prediction for linear dynamical systems. For a survey of recent

literature see (Hazan & Singh, 2022). These techniques make SSMs attractive for sequence tasks which inherently depend on long contexts that scale poorly for transformers. Examples include large language models (Dao et al., 2022), modelling time series (Zhang et al., 2023), and audio generation (Goel et al., 2022). To understand the factors affecting the *memory* in an SSM or simply a linear dynamical system, we now proceed to delineate how past states and inputs affect the future.

**Geometric decay in LDS.** The linear equations governing the dynamics are recursive in nature, and imply that in a noiseless environment, the $t$'th output can be written as

$$y_t = Cx_t + Du_t = C(Ax_{t-1} + Bu_t) + Du_t = ... = \sum_{i=0}^{t-1} CA^i Bu_{t-i} + Du_t$$

The matrix $A$ is asymmetric in general, and can have complex eigenvalues. If the amplitude of these eigenvalues is $> 1$, then the output $y_t$ can grow without bounds. This is called an "explosive" system. In a well-behaved system, the eigenvalues of $A$ have magnitude $\leq 1$. If the magnitudes are bounded away from 1, say $|\lambda_i(A)| \leq 1 - \delta$, for some $\delta > 0$ (referred to as spectral gap), then we can write

$$y_t = \sum_{i=0}^{l} CA^i Bu_{t-i} + \omega_l \quad, \quad \|\omega_l\| \leq \epsilon$$

for $l = O(\frac{1}{\delta} \log \frac{1}{\epsilon})$. This mathematical fact implies that the **effective memory** $l$ of the system is on the order of $\frac{1}{\delta}$, where the input affects the system minimally after $l$ time steps. In general, the parameter $\delta$ is unknown apriori and can get arbitrarily small as we approach systems with have long range dependencies leading to instability in training linear dynamical systems with a long context. This issue is specifically highlighted in the work of Orvieto et al. (2023) who observe that on long range tasks learning an LDS directly does not succeed and requires interventions such as stable exponential parameterizations and specific normalization which have been repeatedly used either implicitly or explicitly in the SSM literature (Gu et al., 2021a). Unfortunately these reparametrizations and normalizations come with no theoretical guarantees. In fact this limitation is generally known to be fundamental to the use of linear dynamical systems, and can only be circumvented via a significant increase in sample complexity (Ghai et al., 2020) or via control over the input sequence (Simchowitz et al., 2018).

**Spectral filtering for linear dynamical systems.** A notable deviation from the standard theory of linear dynamical systems that allows efficient learning in the presence of arbitrarily long memory is the technique of spectral filtering (Hazan et al., 2017). The idea is to project the sequence of inputs to a small subspace that is constructed using special structure of discrete LDS where successive powers of the system matrix appear in the impulse response function. The basic idea is to represent the output as

$$y_t = \sum_{j=1}^{k} M_j \left( \sum_i \phi_j(i) \cdot u_{t-i} \right),$$

where $\phi_j$ are *spectral filters* which are sequence-length sized vectors that given the target sequence length can be computed offline, and $M_j$ are matrices parameterizing the model. These spectral-filters are the eigenvectors of the matrix constructed as the average of outer products of the discrete impulse-response functions, viz $Z = \int_0^1 [1, \alpha, \alpha^2 ...][1, \alpha, \alpha^2 ...]^\top d\alpha$. It is shown that this matrix is inherently low-dimensional and for all $\alpha \in [0, 1]$, vectors of the form $[1, \alpha, \alpha^2 \ldots]$ are well approximated by the top-eigenspace of Z. Figure 1 depicts these filters. For the details of how these filters are derived and their computation, see Section 2.

**Why is spectral filtering important?** The main advantages of spectral filtering are representational efficiency and stability of learning. Consider the naive parameterization of learning a mapping from the system input to output. If we attempt to learn the dynamics matrices $A, B, C, D$ explicitly, the problem becomes non-convex due to the terms $CA^i B$. On the other hand, if we use improper learning and learn the linear map from $u_{t-i}$ to $y_t$ separately for each $i$, then $\frac{1}{\delta}$ matrices are required to express this mapping where $\frac{1}{\delta}$ is the effective memory of the system. With spectral filtering, for linear dynamical systems with

symmetric matrices $A$, the number of filters (and thus the number of matrix parameters) required to represent an observation is **independent of the spectral gap parameter $\delta$!** Moreover, these parameters can be learned stably via regression, as spectral filtering is a convex relaxation of the original non-convex problem. These guarantees indicate that if we featurize the input into the spectral basis, we can potentially design models that are capable of efficiently and stably representing systems with extremely long memory even with $\delta \to 0$. These striking properties motivate our derivation of the recurrent spectral architecture, and is the underlying justification for the performance and training stability gains we see in experiments.

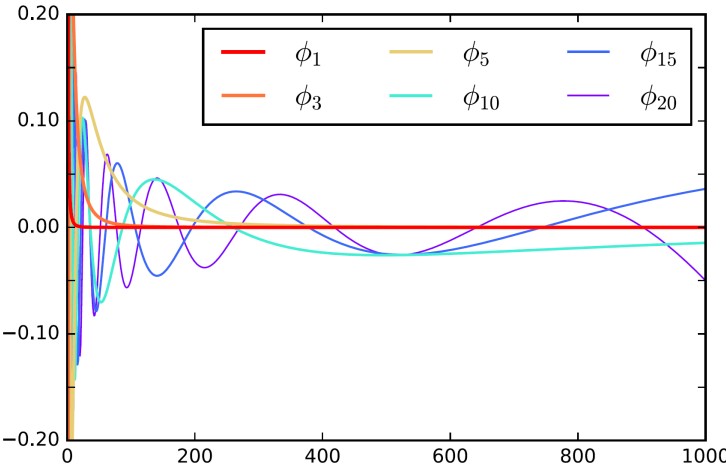

Figure 1: Spectral Filters used by the Spectral Filtering Algorithm. The x-axis is the time domain.

## 1.1 Our Contributions

We start by proposing state space models with learned components that apply spectral filtering for their featurization. We consider two types of spectral filters, which augment the original spectral filters proposed in Hazan et al. (2017) with negative eigenvalues in two different ways. Our main contribution is a neural architecture that is based on these spectral state space models. This neural architecture can be applied recursively in layers, resulting in an expressive architecture for modeling sequential data.

Finally we implement this neural architecture and apply it towards synthetically generated data as well as the Long Range Arena benchmark (Tay et al., 2021). We demonstrate that spectral state space models can stably and more efficiently learn on sequence modelling tasks with long range dependencies without the need for exponential parameterizations, particular initializations and normalizations.

**Main Advantages of Spectral SSM.** Previously proposed convolutional models for sequence modeling, surveyed in the related work section, learn the kernels from the data. The kernels used in Spectral SSM are **theoretically-founded and fixed** and thus parameter-free. In addition, our models are **provably as expressive** as an LDS. In particular, their expressiveness neither depends on the spectral gap nor on the dimension of the system, which are necessary in all other methods.

## 1.2 Related work

Due to limited space, we provide a short overview of the most related work to us below and provide a detailed report on the related work in the appendix (Section A).

**State space models.** SSMs for learning long range phenomenon have received much attention in the deep learning community in recent years starting with the works (Gu et al., 2020),(Gu et al., 2021b) which propose and develop the HiPPO theory. Gu et al. (2021a) develop the S4 parameterization to address the bottlenecks of training efficiency, performance and numberical stability. The S4 parameterization restricts

the system matrices $A$ to be normal plus low-rank, allowing for stable diagonalization. The S4 model was further streamlined in later works, viz. using diagonal system matrices without a loss in performance (Gupta et al., 2022) and the S5 model (Smith et al., 2023) which uses a MIMO diagonal system and associative scans for computational efficiency. Orvieto et al. (2023) investigate whether simpler deep Linear Recurrent Units (LRU) can recover the performance of deep SSMs, and provide an affirmative answer under the crucial caveat that specific modifications on linear RNNs, namely the stable exponential parameterization, $\gamma$- normalization and ring initialization, are necessary to learn on certain challenging long-context modeling tasks. We discuss the details of this ablation in the appendix (Section A.1).

**Spectral filtering.** The technique of spectral filtering (Hazan et al., 2017) was developed as a convex improper learning alternative to directly parameterizing an LDS (as in the case of SSMs) leading to an efficient, polynomial-time algorithm and near-optimal regret guarantees. Different from regression-based methods (eg. SSMs) that aim to identify the system dynamics, spectral filtering's guarantee does not depend on the stability of the underlying system, and is the first method to obtain condition number-free regret guarantees for the MIMO setting. Extension to asymmetric dynamical systems was further studied in Hazan et al. (2018).

**Convolutional Models for Sequence Modeling.** Exploiting the connnection between LDS and convolutions (Gu et al., 2021a), various convolutional models have been proposed for sequence modelling. (Fu et al., 2023) employ direct learning of convolutional kernels but find that they underperform SSMs, identifying non-smoothness of kernels to be the culprit and propose applying explicit smoothing and squashing operations. (Li et al., 2022) identifies two key characteristics of convolutions to be crucial for long range modelling, decay in filters and small number of parameters parameterizing the kernel. (Shi et al., 2023) propose a multiresolution kernel structure inspired from the wavelet transform and multiresolution analysis.

All these methods parameterize the kernels with specific structures and/or add further regularizations to emulate the convolution kernels implied by SSMs. In contrast our proposed kernels are *fixed and thereby parameter-free* and the number of parameters scale in the number of kernels and not the size of the kernel. Furthermore our kernels are *provably at least as expressive as* linear dynamical systems capable of directly capturing and improving the performance of SSMs without the need for specific initializations. Naturally our kernels (see Fig 1) by default satisfy both the smoothness and the decay condition identified (and explicitly enforced) by Li et al. (2022) and Fu et al. (2023).

## 2 Preliminaries

**Sequence prediction.** We treat sequence prediction as a game between a predictor/learner and nature in which iteratively at every time $t \in [L]$, the learner is presented an input $u_t \in \mathbb{R}^{d_{in}}$. The learner $A$ then produces a candidate output $\hat{y}_t = \hat{y}_t(A)$ , and nature reveals the $t^{th}$ element of a target sequence $y_t \in \mathbb{R}^{d_{out}}$. The learner then suffers an instantaneous loss of $\|y_t - \hat{y}_t\|^2$. The task of the learner is to minimize regret over a benchmark set of learning algorithms $\mathcal{A}$, defined as follows

$$\text{Regret} = \sum_{t=1}^{L} \|y_t - \hat{y}_t\|^2 - \min_{A \in \mathcal{A}} \sum_{t=1}^{L} \|y_t - \hat{y}_t(A)\|^2.$$

**Linear Dynamical Systems (LDS):** An example benchmark set of methods is that of a linear dynamical system, which has four matrix parameters, $A \in \mathbb{R}^{N \times N}, B \in \mathbb{R}^{N \times d_{in}}, C \in \mathbb{R}^{d_{out} \times N}, D \in \mathbb{R}^{d_{out} \times d_{in}}$. The system evolves and generates outputs according to the following equations

$$x_t \triangleq Ax_{t-1} + Bu_t, \qquad \hat{y}_t \triangleq Cx_t + Du_t \tag{1}$$

Thus, an example class of benchmark algorithms $\mathcal{A}$ are all predictors that generate $\hat{y}_t$ according to these rules, for a fixed set of matrices $A, B, C, D$.

**Spectral Filtering:** Another important set of predictors is one which is inspired by spectral filtering (Hazan et al., 2017). The spectral filtering theory builds an efficient representation for all vectors in the range

of the function $\mu : [0, 1] \to \mathbb{R}^L$ defined as $\mu(\alpha) \triangleq (\alpha - 1)[1, \alpha, \alpha^2 \ldots]$. To build this representation, for any $L$ define the following Hankel matrix $Z \in \mathbb{R}^{L \times L}$ whose entries are given by

$$Z[i, j] \triangleq \frac{2}{(i + j)^3 - (i + j)}$$

It is shown in the appendix (see Lemma C.1) that $Z = \int_0^1 \mu(\alpha) \mu(\alpha)^\top d\alpha$. Thus it can be seen that $Z$ is a real PSD Hankel matrix. It is known (see Lemma C.4 in the appendix) that real PSD Hankel matrices have an exponentially decaying spectrum. As a result, the crux of the spectral filtering theory, lies in showing that for all $\alpha \in [0, 1]$ [1], the vector $\mu(\alpha)$ is approximately contained in the subspace spanned by the top eigenvectors of Z, making the subspace spanned by top-eigenvectors of Z a very efficient subspace to project the input into. This fact is formalized as Lemma C.3 in the appendix. We now use this intuition to describe the Spectral Filtering algorithm.

Since Z is a real PSD matrix, it admits a real spectral decomposition, and the (non-negative) eigenvalues can be easily ordered naturally by their value. Let $\{(\sigma_j \in \mathbb{R}, \phi_j \in \mathbb{R}^T)\}_{j=1}^L$ be the eigenvalue-eigenvector pairs of $Z$ ordered to satisfy $\sigma_1 \geq \sigma_2 \geq \ldots \geq \sigma_d$. We consider a fixed number $K$ of the above eigenvectors. Algorithms in the spectral filtering class generate $\hat{y}_t$ as follows. For each $k \in K$, we first featurize the input sequence by *projecting* the input sequence until time $t$ on $\phi_k$, leading to a sequence $U_{t,k} \in \mathbb{R}^{d_{in}}$ defined as

$$U_{t,k} = \sum_{i=1}^t u_{t-i} \cdot \phi_k(i).$$

The spectral filtering class is further parameterized by matrices $M_1^u \in \mathbb{R}^{d_{out} \times d_{in}}$, $M_2^u \in \mathbb{R}^{d_{out} \times d_{in}}$ and a set of matrices $M_1^\phi, ..., M_K^\phi \in \mathbb{R}^{d_{out} \times d_{in}}$. The output at time $t$ is then given by

$$\hat{y}_t = \hat{y}_{t-1} + M_1^u u_t + M_2^u u_{t-1} + \sum_{k=1}^K M_k^\phi U_{t,k}. \tag{2}$$

Note that given an input sequence $u_{1:L}$ for any $k$, the $d_{in} \times T$ matrix $U_{1:L,k}$ can be efficiently computed via convolutions along the time dimension $L$ in total time $O(d_{in} \cdot L \log(L))$. The following theorem (proved in Hazan et al. (2017)) establishes that the spectral filtering class of predictors approximately contains bounded linear dynamical systems with positive semi-definite $A$. Here we denote by $\|W\|_{col}$ the maximum column norm of a matrix $W$, and by $y_t^{LDS}$ and $y_t^{SF}$ the outputs of the LDS and Spectral Filtering (via equation 1 and equation 2), respectively.

**Theorem 2.1.** *Consider any linear dynamical system with a PSD transition matrix $A$, such that $\|A\| \leq 1$, and any sequence $u_{1:L}$ satisfying $\|u_t\| \leq a$. Then there exists a set of matrices $M_1^u, M_2^u, M_1^\phi, ..., M_K^\phi$ s.t. Spectral Filtering can approximate the output of the LDS with the following guarantee:*

$$\|y_t^{\text{LDS}} - y_t^{\text{SF}}\|^2 \leq c \cdot \|B\|_{col} \cdot \|C\|_{col} \cdot L^3 \cdot a \cdot e^{-\left(\frac{\pi^2}{4} \cdot \frac{K}{\log(L)}\right)}$$

*where $c \leq 10^6$ is a universal constant.*

We do not provide a proof for this theorem which can be found in Hazan et al. (2017) [2]. Instead, in the next section we provide a generalization of this theory to cover all symmetric matrices and not just PSD matrices and prove a more general theorem (Theorem 3.1). We further build upon this generalization to create a sequence to sequence prediction unit.

## 3 Spectral Transform Unit (STU)

In this section we use Spectral Filtering to create a sequence to sequence neural network layer, i.e. given an input sequence $\{u_1 \ldots u_L\} \in \mathbb{R}^{d_{in}}$, it produces an output sequence $\{y_1 \ldots y_L\} \in \mathbb{R}^{d_{out}}$.

---

[1] in particular all $\alpha$ close to 1, representing marginally stable systems.

[2] Note that Hazan et al. (2017) consider a simpler setting where in the ground truth $y_t$ is available to the learner for all future time steps. We do not make such an assumption and theorems have been adjusted to suffer an additional $L$ factor in the error as a result.

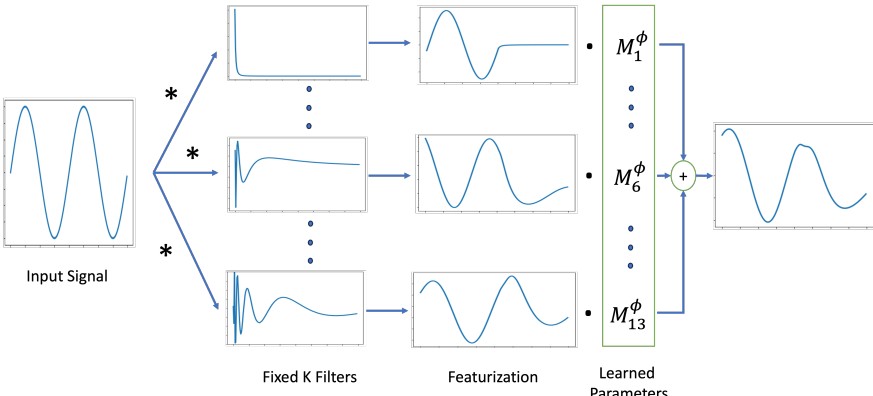

Figure 2: Schematic showing the spectral projection of a 1-dimensional input sequence and how these features are used to produce the spectral component in the STU output equation 4. In the multi-dimensional case the operation is applied in parallel across every input dimension.

A single layer of STU (depicted in Figure 2) is parameterized by a number $K$, denoting the number of eigenvectors and matrices $M_1^{\phi+} \ldots M_K^{\phi+}, M_1^{\phi-} \ldots M_K^{\phi-} \in \mathbb{R}^{d_{\text{out}} \times d_{\text{in}}}$, and $M_1^u, M_2^u, M_3^u \in \mathbb{R}^{d_{\text{out}} \times d_{\text{in}}}$. The matrices form the *params* of the layer. Further recall the Hankel matrix $Z \in \mathbb{R}^{L \times L}$ whose entries are given by

$$Z[i,j] \triangleq \frac{2}{(i+j)^3 - (i+j)}. \tag{3}$$

and let $\{(\sigma_j \in \mathbb{R}, \phi_j \in \mathbb{R}^L)\}_{j=1}^L$ be the eigenvalue-eigenvector pairs of $Z$ ordered to satisfy $\sigma_1 \geq \sigma_2 \geq \ldots \sigma_d$. Given an input sequence $\{u_1 \ldots u_L\} \in \mathbb{R}^{d_{\text{in}}}$, we first featurize the input sequence as follows. For any $t, k$, we begin by *projecting* the input sequence till time $t$ on *fixed* filters $\phi_k$, leading to two feature vectors $U_{t,k}^+, U_{t,k}^- \in \mathbb{R}^{d_{\text{in}}}$ defined as

$$U_{t,k}^+ = \sum_{i=0}^{t-1} u_{t-i} \cdot \phi_k(i) \qquad U_{t,k}^- = \sum_{i=0}^{t-1} u_{t-i} \cdot (-1)^i \cdot \phi_k(i).$$

Note that for every $k$, the sequence of features $U_{1:L,k}$ can be computed efficiently via convolution. The output sequence $\{y_1 \cdots y_L\}$ is then given by

$$\hat{y}_t = \underbrace{\hat{y}_{t-2} + \sum_{i=1}^3 M_i^u u_{t+1-i}}_{\text{Auto-regressive Component}} + \underbrace{\sum_{k=1}^K M_k^{\phi+} \sigma_k^{1/4} U_{t-2,k}^+ + \sum_{k=1}^K M_k^{\phi-} \sigma_k^{1/4} U_{t-2,k}^-}_{\text{Spectral Component}}. \tag{4}$$

The above output contains a small auto-regressive component that essentially allows for stable learning of the spectral component as the memory grows. The differences from the original spectral filtering class equation 2 are the introduction of a *negative* part in the spectral component and the slight change in the auto-regressive component. Both of these changes are necessitated by the requirement to capture negative eigenvalues of $A$. Note that equation 4 corresponds to the specification of the algorithm presented in Hazan et al. (2018), when the eigenvalues are known to be real numbers. For completeness and ease of discourse we prove the following representation theorem in the Appendix which shows that the above class approximately contains any marginally-stable LDS with symmetric $A$.[3] As before, let $\|M\|_{\text{col}}$ denote the maximum column norm of a matrix $M$.

**Theorem 3.1.** *Consider any linear dynamical system with a symmetric matrix $A$ where $\|A\| \leq 1$, and any sequence $u_{1:L}$ satisfying $\|u_t\| \leq a$. Then there exists matrices $M_1^u, M_2^u, M_3^u, M_1^{\phi+} \ldots M_K^{\phi+}, M_1^{\phi-} \ldots M_K^{\phi-} \in$*

---

[3]We discovered some small but easily fixable errors in the original proof of Hazan et al. (2017) which we have corrected in our proof

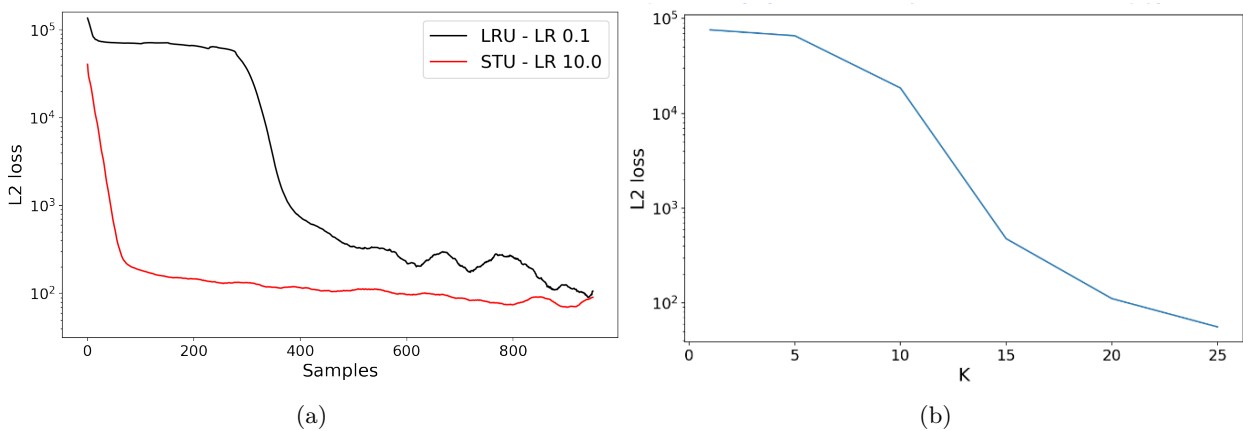

(a)                                      (b)

Figure 3: Learning dynamics for learning a marginally stable LDS. (a.)(Smoothed) Learning curves for a single STU layer (red) vs a single LRU layer (black). The learning rate was tuned for both models. See Appendix for a detailed discussion of the tuning and sensitivity to hyperparameters for both the models. Curiously at stable LRs we observe that LRUs show a plateauing of learning. (b.) Error (in log-scale) obtained by the single STU layer as a function of the model parameter 'K'. We observe an exponential drop in the reconstruction loss as predicted by the analysis.

$\mathbb{R}^{d_{\text{out}} \times d_{\text{in}}}$, such the output of the LDS (via equation 1) can be approximated by Spectral Filtering (via equation 4) with the following guarantee:

$$\|y_t^{\text{LDS}} - y_t^{\text{SF}}\|^2 \le c \cdot \|B\|_{col} \cdot \|C\|_{col} \cdot L^3 \cdot a \cdot e^{-\left(\frac{\pi^2}{4} \cdot \frac{K}{\log(L)}\right)}$$

where $c \le 2 \times 10^6$ is a universal constant.

The above theorem in particular ensures for any sequence length $L$ that setting $K = O\left(\log(L) \log\left(\frac{\|B\|_{\text{col}} \cdot \|C\|_{\text{col}} \cdot L \cdot a}{\epsilon}\right)\right)$ we get there exists a spectral filtering model with $K$ filters that can approximate any LDS upto an error of $\epsilon$. Note that the requirement on the number of filters grows logarithmically in $L$, highlighting the efficiency of the representation. The proof of the above theorem is provided in the appendix along with an alternative version of spectral filtering using slightly modified filters which also provide the same guarantee.

**Remark 3.2.** *Comparing Theorem 3.1 (our contribution) and Theorem 2.1 (Theorem 1 from Hazan et al. (2017)), we note firstly that our theorem holds for symmetric matrices and not just PSD matrices. Hazan et al. (2017) allude to a direct extension for the symmetric case which we believe is not fully correct. We use a similar idea to prove this theorem. Secondly a minor difference is that in the sequential prediction setting the prediction is auto-regressive, i.e. uses its own y to make the future predictions.*

Due to space limitations, we discuss the runtime scaling of our method and compare it with different methods in the appendix (Section A).

### 3.1 Experiment: Learning a marginally-stable LDS

We provide a simple synthetic evaluation of the stability and training efficiency afforded by the STU. We consider a low-dimensional linear system $A, B, C, D$ generated as follows. $B \in \mathbb{R}^{4 \times 3}, C \in \mathbb{R}^{3 \times 4}$ are matrices with iid unit Gaussian entries. $D$ is a diagonal matrix with iid unit Gaussian entries and $A$ is a diagonal matrix with $A_{ii} \sim 0.9999 * Z$ where $Z$ is a random sign. By design this is a system with a very high stability constant ($\sim 10^4$). As a training dataset we generated $\{(u_i, y_i)\}$ where $u_i$ is a random input sequence and $y_i$ is the output generated by applying the linear dynamical system on $u_i$. We perform mini-batch (batch size 1) training with the l2 loss. As comparison we perform the same procedure with an LRU (Linear Recurrent Unit) layer as proposed by Orvieto et al. (2023) which directly parameterizes the linear system. The results of the training loss as seen by the two systems are presented in Figure 3a.

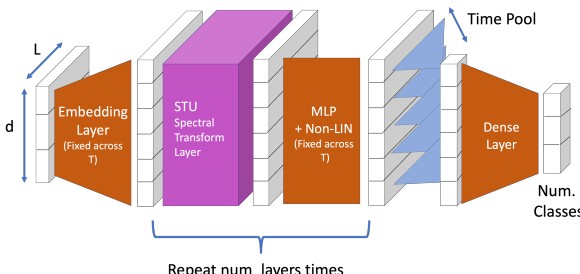

(a) Schematic displaying a multi-layer STU model.

| Model | Specification | Pathfinder | PathX |
|-------|---------------|------------|-------|
| STU | Eqn equation 4 (K=16) | 91.8 | 89.5 |
| LRU | Dense A | ✗ | ✗ |
|  | $\Lambda$ Exp. Param. | 65.4 | ✗ |
|  | $\Lambda$ Stable Exp. | 93.5 | ✗ |
|  | + Ring Init. | 94.4 | ✗ |
|  | + $\gamma$-Norm. | 95.1 | 94.2 |

(b) Comparison of the basic stacked STU model against LRU ablations in Orvieto et al. (2023)

We use all the initialization/normalization techniques as recommended by Orvieto et al. (2023) for LRU including the *stable exponential parameterization*, *$\gamma$-normalization* and *ring-initialization*. Indeed we find that all these tricks were necessary to learn this system at all. We provide more details about the ablations and other hyperparameter setups in the appendix. We observe that the STU is significantly more efficient at learning the LDS as opposed to the LRU. We further find that there is a wide range of LRs where the STU has a stable optimization trajectory and the loss decreases continuously highlighting the advantages of a convex parameterization. On the other hand, LRU is able to eventually learn the system at the right learning rates, it requires almost 8x the number of samples to get to a system with non-trivial accuracy. More details can be found in the appendix. Curiously we observe that for the LRU training plateaus completely for the first 50% of training highlighting the difficulty of optimization via a non-convex landscape.

The STU layer in the previous experiment employs $K = 25$. In Figure 3b we plot the performance of STU at various levels of $K$. As predicted by the theory we observe an exponential decay in the error as $K$ increases with the error effectively plateauing after $K \geq 15$.

## 4 Stacked STU

To increase the representation capacity and to maintain the efficiency of prediction through linear dynamical systems, proposed models in the SSM literature take the form of stacking these sequence to sequence transforms into multiple layers. Non-linearities in the model can then be introduced by sandwiching them as layers lying in between these sequence to sequence transforms.

In this paper we closely follow the stacking approach followed by Orvieto et al. (2023), replacing the LRU layers appropriately by STU layers. A schematic for the resultant multi-layer model is displayed in Figure 4a. In a nutshell, the input sequence is first embedded via a time-invariant embedding function followed by multiple repetitions of alternating STU layers and non-linearities (in particular we use GLU). Finally the resulting output is time-pooled followed by a final readout layer according to the task at hand. This composite model can now be trained in a standard fashion via back-propagation and other commonly used deep-learning optimization techniques.

### 4.1 Experiments on Long Range Arena (Tay et al., 2021)

We evaluate the stacked STU model on the Long Range Arena (LRA) benchmark (Tay et al., 2021). This benchmark aims to assess the performance of sequence prediction models in long-context scenarios and consists of six tasks of various modalities, including text and images. The context length for the tasks ranges from 1K to 16K, and the tasks require capabilities such as hierarchical reasoning, matching and retrieval, and visual-spatial understanding. SSMs (Gu et al., 2021a) have shown significantly superior performance on most of the tasks compared to Transformer architectures. In particular for the hardest task in the suite, PathX (image classification with context length of 16K), no transformer model has been able to achieve accuracy beyond random guessing. We provide the evaluation of the stacked STU model on the two hardest tasks namely PathFinder and PathX in Table 4b.

| | CIFAR | ListOps | Text | Retrieval | Pathfinder | PathX |
|---|---|---|---|---|---|---|
| S4 (Gu et al., 2021a) | 88.65 | 59.60 | 86.82 | 90.90 | 94.20 | 96.35 |
| LRU (Orvieto et al., 2023) | 89 | 60.2 | 89.4 | 89.9 | 95.1 | 94.2 |
| AR-STU | **91.34** | **61.14** | **90.47** | 90.52 | **95.45** | 93.24 |

Table 1: Comparison of the STU model against various proposed SSM models on the LRA benchmark. We report the median over 5 trials for our experiments.

We compare our performance against the ablation carried out by Orvieto et al. (2023) who find that ring initialization, stable exponential parameterization and $\gamma$-normalization are all crucial towards learning these tasks. In particular as reported by Orvieto et al. (2023) all three of the above interventions were necessary to learn on PathX to any non-trivial accuracy. This is a result of the much larger context length of 16K employed by the PathX task. On the other hand we find that the the stacked STU (with the STU component exactly as represented by equation 4) is sufficient to learn on both these tasks to relatively high accuracies. Notably we do not require any other normalizations or initialization techniques We initialize all the parameters of the STU i.e. M matrices to 0. Details about our implementation as well as details about the experiments including hyperparameters can be found in the appendix (Section E. This result in particular confirms and highlights the theoretical stability afforded by the STU even under learning tasks involving large sequence lengths. In the appendix (Table 2 we provide the performance evalaution of the stacked STU on all tasks of the LRA benchmark.

In the next section we highlight a simple technique towards significantly improving the achieved accuracy for the stacked STU model.

## 5 Hybrid Temporal and Spectral Units

A simple extension to the STU model (Equation equation 4) is to parameterize the dependence of $y_t$ on $y_{t-2}$ with a parameter $M_y$, leading to the following prediction model

$$\hat{y}_t = \underbrace{\boldsymbol{M^y}\hat{y}_{t-2} + \sum_{i=1}^{3} M_i^u u_{t+1-i}}_{Auto-regressive\ Component} + \underbrace{\sum_{k=1}^{K} M_k^{\phi+}\sigma_k^{1/4}U_{t-2,k}^{+} + \sum_{k=1}^{K} M_k^{\phi-}\sigma_k^{1/4}U_{t-2,k}^{-}}_{Spectral\ Component}. \tag{5}$$

Setting $M^y = I$ we recover the guarantees afforded by Theorem 3.1 and thus the above model is strictly more powerful. We find that the above change leads to significant improvements over the accuracy achieved by the simple STU model. We can further extend the auto-regression to depend on multiple previous $y$ as opposed to just $y_{t-2}$. Indeed as the following theorem shows adding sufficiently long auto-regression is powerful enough to capture any LDS.

**Theorem 5.1.** *Given an LDS parameterized by $A \in \mathbb{R}^{d \times d}, B, C, D$, there exist coefficients $\alpha_{1:d}$ and matrices $\Gamma_{0:d}$ such that given any input sequence $u_{1:L}$, the output sequence $y_{1:L}$ generated by the action of the LDS on the input satisfies for all $t$*

$$y_t = \sum_{i=1}^{d} \alpha_i y_{t-i} + \sum_{i=0}^{d} \Gamma_i u_{t-i}$$

This is a well-known observation and we provide a proof in the appendix (Section F). Motivated by the above theorem we propose a generalization of STU, which we call AR-STU, to add auto-regression over the

previously produced outputs. In particular given a parameter $k_y$ we define AR-STU as

$$\hat{y}_t = \underbrace{\sum_{i=1}^{k_y} \boldsymbol{M_i^y} \hat{y}_{t-i} + \sum_{i=1}^{3} M_i^u u_{t+1-i}}_{\mathrm{Auto-regressive\ Component}} + \underbrace{\sum_{k=1}^{K} M_k^{\phi+} \sigma_k^{1/4} U_{t-2,k}^+ + \sum_{k=1}^{K} M_k^{\phi-} \sigma_k^{1/4} U_{t-2,k}^-}_{\mathrm{Spectral\ Component}} . \tag{6}$$

In Table 1, we evaluate the performance of AR-STU on Long Range Arena. In our experiments we search over two values of $k_y = \{2, 32\}$. For non-image tasks, ListOps, Text and Retrieval, we find that setting $k_y = 2$ is sufficient to get optimal results. For the image tasks, CIFAR, Pathfinder and PathX, we found that $k_y = 32$ led to significant performance gains. A performance ablation over this parameter can be found in the appendix (Table 2). Overall we find that the STU model provides improvements over baselines such as S4 and LRU on 4 out of the 6 tasks and performs comparably to the best baseline on the others. Remarkably, the STU layers come with provable guarantees and thus performs well *out of the box* without the need for specific initializations, discretizations or normalizations. We initialize all parameters $M_i^y, M_i^u, M_k^{\phi+}, M_k^{\phi-}$ with 0. We provide details of the experimental setup, including hyperparameter tuning in the appendix (Section E).

## 6 Conclusion

Inspired by the success of SSMs, we present a new theoretically-founded deep neural network architecture, Spectral SSM, for sequence modelling based on the Spectral Filtering algorithm for learning Linear Dynamical Systems. The SSM performs a reparameterization of the LDS and is guaranteed to learn even marginally stable symmetric LDS stably and efficiently. We demonstrate the core advantages of the Spectal SSM, viz. robustness to long memory through experiments on a synthetic LDS and the Long Range Arena benchmark. We find that the Spectral SSM is able to learn even in the presence of large context lengths/memory without the need for designing specific initializations, discretizations or normalizations which were necessary for existing SSMs to learn in such settings. While spectral SSMs only model symmetric A, our presented set of experiments on the LRA benchmark suggest that the gap between symmetric and general A is potentially small in real world tasks. Indeed more recent SSM models like Gu & Dao (2023); De et al. (2024) work with real diagonals (i.e. symmetric case) as they do not find evidence that adding complex eigenvalues help. Spectral filtering has been extended in certain settings to asymmetric A (Hazan et al., 2018) and a similar extension to our proposal is straightforward but comes with efficiency losses and we leave it to future work.

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

## A   Detailed Related work

**State space models.**   SSMs for learning long range phenomenon have received much attention in the deep learning community in recent years. Gu et al. (2020) propose the HiPPO framework for continuous-time memorization, and shows that with a special class of system matrices $A$ (HiPPO matrices), SSMs have the capacity for long-range memory. Subsequently, Gu et al. (2021b) propose the Linear State-Space Layer (LSSL), where the system matrix is learnable. The LSSL can be viewed as a recurrence in the state domain and a convolution in the time domain, and generalizes particular RNN and CNN architectures. For efficient learning of the system matrices, authors propose learning within a class of structured matrices that contain the HiPPO dynamics, and have efficient convolution schemes. However, the proposed method is numerically unstable in practice as well as memory-intensive. As a result, Gu et al. (2021a) develop the S4 parameterization to address these bottlenecks. The S4 parameterization restricts the system matrices $A$ to be normal plus low-rank, allowing for stable diagonalization of the dynamics. Under this parameterization, authors design memory and computationally efficient methods that are also numerically stable.

The S4 model has been further streamlined in later works. Gupta et al. (2022) simplify the S4 parameterization to diagonal system matrices, and shows that the diagonal state-space model (DSS) is competitive with S4 on several benchmarks. Smith et al. (2023) propose the S5 architecture, which improves upon S4 in two

directions: 1) instead of having independent SISO SSMs in the feature dimension, S5 has one MIMO DSS that produces vector-valued outputs; 2) S5 uses efficient parallel scans in place of convolutions, bypassing custom-designed algorithms for computing the convolutional filters.

To improve the performance of SSMs on language modeling tasks, Dao et al. (2022) develops the H3 layer by stacking two SSMs together. They identify two areas where SSMs underperform compared to the transformer: remembering earlier tokens and comparing tokens across the input sequence. The H3 layer includes a shift SSM, where the dynamics matrix is a shifting operator, and a DSS, with multiplicative interactions. The shift SSM enables the layer to store earlier tokens, while the multiplicative interaction allows for comparison (inner product) between tokens in a sequence. They also develop FFT algorithms with better hardware utilization, to close the speed gap between SSMs and Transformers.

Motivated by the similarities between SSMs and RNNs, Orvieto et al. (2023) investigate whether deep RNNs can recover the performance of deep SSMs, and provide an affirmative answer. The proposed RNN architecture is a deep model with stacked Linear Recurrent Unit (LRU) layers. Each LRU has linear recurrence specified by a complex diagonal matrix, learned with exponential parameterization and proper normalization techniques. The deep LRU architecture has comparable computational efficiency as SSMs and matches their performance on benchmarks that require long-term memory. However, the paper also shows that without the specific modifications on linear RNNS, namely the stable exponential parameterization, gamma normalization and ring initialization, LRU fails to learn on certain challenging long-context modeling tasks. We provide further details about this study after this section.

**Spectral filtering.** The technique of spectral filtering for learning linear dynamical systems was put forth in Hazan et al. (2017). This work studies online prediction of the sequence of observations $y_t$, and the goal is to predict as well as the best symmetric LDS using past inputs and observations. Directly learning the dynamics is a non-convex optimization problem, and spectral filtering is developed as an improper learning technique with an efficient, polynomial-time algorithm and near-optimal regret guarantees. Different from regression-based methods that aim to identify the system dynamics, spectral filtering's guarantee does not depend on the stability of the underlying system, and is the first method to obtain condition number-free regret guarantees for the MIMO setting. Extension to asymmetric dynamical systems was further studied in Hazan et al. (2018).

**Convolutional Models for Sequence Modeling** Exploiting the connnection between Linear dynamical systems and convolutions (as highlighted by Gu et al. (2021a)) various convolutional models have been proposed for sequence modelling. Fu et al. (2023) employ direct learning of convolutional kernels directly to sequence modelling but find that they underperform SSMs. They find the non-smoothness of kernels to be the culprit and propose applying explicit smoothing and squashing operations to the kernels to match performance on the Long Range Arena benchmark. The proposed model still contains significantly large number of parameters growing with the sequence length. Li et al. (2022) identifies two key characteristics of convolutions to be crucial for long range modelling, decay in filters and small number of parameters parameterizing the kernel. They achieve this via a specific form of the kernel derived by repeating and scaling the kernel in a dyadic fashion. Shi et al. (2023) propose a multiresolution kernel structure inspired from the wavelet transform and multiresolution analysis.

All these methods parameterize the kernels with specific structures and/or add further regularizations to emulate the convolution kernels implied by SSMs. In contrast our proposed kernels are *fixed and thereby parameter-free* and the number of parameters scale in the number of kernels and not the size of the kernel. Furthermore our kernels are *provably more expressive* than linear dynamical systems capable of directly capturing and improving the performance of SSMs without the need for specific initializations. Naturally our kernels (see Fig 1) by default satisfy both the smoothness and the decay condition identified (and explicitly enforced) by Li et al. (2022) and Fu et al. (2023).

## A.1 Ablations performed by Orvieto et al. (2023)

Motivated by the success of SSMs, Orvieto et al. (2023) revisit the RNN model (under the same deep stacked structure as SSMs) to investigate their efficiency. They begin from a simple linear RNN (a directly

parameterized LDS) and add multiple components inspired from the SSM literature to ensure numerical stability and trainability of the model especially as the sequences grow larger. Overall they demonstrate that carefully designed parameterizations and initializations of LDS parameters as well as specifically designed normalizations are all necessary for model to learn consistently over the LRA dataset and in particularly over the 16K context length task PathX. These interventions are driven by specific intuitions such as an inductive bias towards larger memory or controlling the loss blowup at initialization under long contexts but as such come with no theoretical guarantees towards alleviating the problem. We provide some quick details towards what these interventions are and refer the reader to Orvieto et al. (2023) to understand the motivations behind them and comparisons with similar ideas existing in previous SSM literature. The LRU model considered by Orvieto et al. (2023) is given by

$$y_k = \text{diag}(\lambda)y_{k-1} + \gamma \odot Bu_k.$$

In the above the learned parameters are $\lambda$ and B and note that $\text{diag}(\lambda)$ corresponds to a diagonal A. $\gamma$ is a specific normalization technique they develop to control the loss blowup under long-context detailed below. They perform the following interventions towards stable training

- **Stable Exponential Parameterization**: They parameterize $\lambda$ as

$$\lambda_j = \underbrace{\exp(-\exp(\nu_j^{\log}))}_{\text{magnitude}} + i\underbrace{\exp(\theta_j^{\log}))}_{\text{phase}}$$

  The above is done to ensure a bound on the magnitude of eigenvalues of the effective A matrix as well as to ensure more *resolution* in the parameter space closer to the value of 1.

- **Ring Initialization**: They initialize the $\lambda_j$ in the complex annulus [min_rad, max_rad]. This ensures that at initialization the magnitude of $\lambda_j$ chosen randomly lies in $\in$ [min_rad, max_rad] and the phase is chosen randomly. When not applying this intervention min_rad and max_rad are chosen to be 0,1 respectively. When applying this intervention these values are chosen to be closer to 1, e.g. $0.9, 0.999$ respectively.

- **$\gamma$-Normalization**: They set $\gamma_j = \sqrt{1 - |\lambda_j|^2}$

- **Restricting Phase at initialization**: Instead of drawing a random phase at initialization the authors recommend selecting the initial phase from $[0, \pi/10]$. The authors claim that uniform phase inherently biases the network towards learning spurious features in the input sequence.

Orvieto et al. (2023) provide the following ablation in the paper. In particular we see that all the above interventions are necessary to make the model get to non-trivial accuracy on PathX. On the contrary, as we show the STU model achieves comparable accuracy without requiring any specific initialization or normalization.

| Model | Specification | sCIFAR | ListOps | Pathfinder | PathX |
|-------|---------------|--------|---------|------------|-------|
| LRU | Dense A | 72.2 | 50.4 | ✗ | ✗ |
| | $\Lambda$ Exp. Param. | 85.4 | 60.5 | 65.4 | ✗ |
| | $\Lambda$ Stable Exp. Param. | 87.2 | 59.4 | 93.5 | ✗ |
| | + Ring Init. | 88.1 | 59.4 | 94.4 | ✗ |
| | + $\gamma$-Norm. + Phase Init. | 89.0 | 60.2 | 95.1 | 94.2 |

## B    Computational complexity and comparison to other methods.

Using the STU method to make a sequence of $L$ predictions, the features $U^+, U^- \in \mathbb{R}^{L \times d_{\text{in}} \times K}$ can be computed in time $O(K \cdot L \cdot d_{\text{in}} \log(L))$ using the Discrete Fast Fourier Transform, where $K$ is the number of

filters and $L$ is the context length. The linear prediction part (i.e. spectral component) takes $O(K \cdot L \cdot d_{\text{in}} \cdot d_{\text{out}})$ time, and the autoregressive part can be implemented in total time $O(L \cdot d_{\text{in}} \cdot d_{\text{out}})$. Therefore the overall runtime is $O(K \cdot L \cdot d_{\text{in}} \cdot (\log(L) + d_{\text{out}}))$. [4]

For comparison, consider LRU and transformers. The same computation carried out by LRU w. diagonal system matrices is dominated by the hidden dimension, i.e. $O(L \cdot d_{\text{hidden}} \cdot (d_{\text{in}} + d_{\text{out}}))$. Thus, the number of filters is replaced by $d_{\text{hidden}}$, which is usually an order of magnitude larger, although STU has another $O(\log L)$ overhead.

A transformer model with full attention runs in time $O(L^2 d_{\text{in}} d_{\text{out}})$, which is significantly more costly than both LRU and STU. This is consistent with the motivation of SSM as more efficient models for sequences.

## C  Proof of Theorem 3.1

We begin by observing that without loss of generality we can assume that $A$ is a real-diagonal matrix. This can be ensured by performing a spectral decomposition of $A = U\Sigma U^\top$ and *absorbing* the $U, U^\top$ by redefining the system. Before continuing with the proof, we will provide some requisite definitions and lemmas. Define the following vector for any $\alpha \in \mathbb{R}$, $\mu(\alpha) \in \mathbb{R}^L$, with $\mu(\alpha)(i) = (\alpha-1)\alpha^{i-1}$. Further define the Hankel matrix $H$ as

$$Z \triangleq \int_0^1 \mu(\alpha)\mu(\alpha)^\top d\alpha.$$

As the following lemma shows the Hankel matrix $Z$ above is the same Hankel matrix defined in the definition of STU equation 3.

**Lemma C.1.** *$Z$ is a Hankel matrix with entries given as*

$$Z(i,j) = \frac{2}{(i+j)^3 - (i+j)}$$

**Lemma C.2.** *We have that the following statements hold regarding $\mu(\alpha)$ for any $\alpha \in [0,1]$,*

- *$|\mu(\alpha)|^2 \le 1$*

- *For any $\alpha \in [0,1]$ and any unit vector $v$ we have that*

$$(\mu(\alpha)^\top v)^2 \le 12(v^\top H v)$$

**Lemma C.3.** *For any $\alpha \in [0,1]$, let $\tilde{\mu}(\alpha)$ be the projection of $\mu(\alpha)$ on the subspace spanned by top $k$ eigenvectors of $Z$, then we have that*

$$\|\mu(\alpha) - \tilde{\mu}(\alpha)\|^2 \le 12 \sum_{i=k+1}^L \sigma_i$$

Finally the following lemma from Hazan et al. (2017) shows that the spectrum of the matrix $Z$ decays exponentially.

**Lemma C.4** (Lemma E.3 Hazan et al. (2017))**.** *Let $\sigma_j$ be the top $j^{th}$ eigenvalue of $Z$. Then we have that*

$$\sigma_j \le \Gamma c^{-j/\log(L)}$$

*where $c = e^{\pi^2/4} \sim 11.79$ and $\Gamma = 235200$ is an absolute constant.*

We now move towards proving Theorem 3.1. Consider the following calculation for the LDS sequence $y_t^{\text{LDS}}$

$$y_t^{\text{LDS}} = \sum_{i=0}^T CA^i Bu_{t-i} + Du_t,$$

---

[4]We shortly note that the $K$ filters can be distributed amongst $K$ machines and their computations done separately. There are many other opportunities for distributed computing for all architectures which we will not survey here as it is out of scope.

and therefore we have that

$$y_t^{\mathrm{LDS}} - y_{t-2}^{\mathrm{LDS}} = (CB + D)u_t + CABu_{t-1} - Du_{t-2} + \underbrace{\sum_{i=0}^{T} C(A^{i+2} - A^i)Bu_{t-2-i}}_{\text{Term of Interest}}$$

For any $t_1 \geq t_2$ we define the matrix $\bar{U}_{\{t_1:t_2\}} \in \mathbb{R}^{d_{\mathrm{out}} \times t_1 - t_2 + 1}$ whose $i^{th}$ column is the input vector $u_{t_1-i+1}$. We allow $t_2$ to be negative and by convention assume $u_t = 0$ for any $t \leq 0$. Denote the diagonal entries of $A$ by $\{\alpha_l\}_{l=1}^{d_h}$, i.e. $\alpha_l = A(l, l)$. Further let $b_l, c_l$ be the $l$-th column for the matrices $B, C$ respectively. The term of interest above can then be written as

$$\sum_{i=0}^{L} C(A^{i+2} - A^i)Bu_{t-2-i}$$

$$= \sum_{l=1}^{d_h} (c_l \otimes b_l) \left( \sum_{i=0}^{L} (\alpha_l^{i+2} - \alpha_l^i)u_{t-2-i} \right)$$

$$= \sum_{l:\alpha_l \geq 0} (c_l \otimes b_l) \left( \sum_{i=0}^{L} (\alpha_l^2 - 1)\alpha_l^i u_{t-2-i} \right) + \sum_{l:\alpha_l < 0} (c_l \otimes b_l) \left( \sum_{i=0}^{L} (\alpha_l^2 - 1)\alpha_l^i u_{t-2-i} \right)$$

$$= \sum_{l:\alpha_l \geq 0} (\alpha_l + 1)(c_l \otimes b_l) \left( \sum_{i=0}^{L} (\alpha_l - 1)\alpha_l^i u_{t-2-i} \right) + \sum_{l:\alpha_l < 0} (1 + |\alpha_l|)(c_l \otimes b_l) \left( \sum_{i=0}^{L} (|\alpha_l| - 1)|\alpha_l|^i (-1)^i u_{t-2-i} \right)$$

$$= \sum_{l:\alpha_l \geq 0} (\alpha_l + 1)(c_l \otimes b_l) \left( \bar{U}_{\{t-2:t-1-L\}}\mu(\alpha) \right) + \sum_{l:\alpha_l < 0} (|\alpha_l| + 1)(c_l \otimes b_l) \left( \bar{U}_{\{t-2:t-1-L\}} \odot \mathbf{1}^{\pm} \right) \mu(|\alpha_l|)$$

where $\mathbf{1}^{\pm} \in \mathbb{R}^{d_{\mathrm{out}} \times L}$ is defined as the matrix whose every row is the alternating sign vector $[1, -1, 1, -1 \ldots])$ and $\odot$ is Hadamard product (i.e. entry-wise multiplication).

$$y_t^{\mathrm{LDS}} - y_{t-2}^{\mathrm{LDS}} = (CB + D)u_t + CABu_{t-1} - Du_{t-2} + \underbrace{\sum_{l:\alpha_l \geq 0} (\alpha_l + 1)(c_l \otimes b_l) \left( \bar{U}_{\{t-2:t-1-L\}}\mu(\alpha) \right)}_{PositivePart}$$

$$+ \underbrace{\sum_{l:\alpha_l < 0} (|\alpha_l| + 1)(c_l \otimes b_l) \left( \bar{U}_{\{t-2:t-1-L\}} \odot \mathbf{1}^{\pm} \right) \mu(|\alpha_l|)}_{NegativePart} \tag{7}$$

Recall that we defined the sequence $\{\sigma_k, \phi_k\}_{k=1}^{L}$ to be the eigenvalue and eigenvector pairs for the Hankel matrix $Z$. For any $\alpha$ we define the projection of $\mu(\alpha)$ on the top $k$ eigenvectors as $\tilde{\mu}(\alpha)$, i.e. $\tilde{\mu}(\alpha) = \sum_{k=1}^{K} (\mu(\alpha_l)^{\top}\phi_k)\phi_k$. Further define STU parameters as follows

$$M_1^u = CB + D, M_2^u = CAB, M_3^u = -D$$

$$M_k^{\phi+} = \sum_{l:\alpha_l \geq 0} (\alpha_l + 1)(\mu(\alpha_l)^{\top}\phi_k)\sigma_k^{-1/4}(c_l \otimes b_l)$$

$$M_k^{\phi-} = \sum_{l:\alpha_l < 0} (|\alpha_l| + 1)(\mu(|\alpha_l|)^{\top}\phi_k)\sigma_k^{-1/4}(c_l \otimes b_l) \tag{8}$$

By the definition of STU prediction equation 4 we have that,

$$y_t^{\mathrm{STU}} = y_{t-2}^{\mathrm{STU}} + \sum_{i=1}^{3} M_i^u u_{t+1-i} + \sum_{k=1}^{K} M_k^{\phi+}\sigma_k^{1/4} \left( \sum_{i=0}^{t-1} u_{t-i} \cdot \phi_k(i) \right) + \sum_{k=1}^{K} M_k^{\phi-}\sigma_k^{1/4} \left( \sum_{i=0}^{t-1} u_{t-i} \cdot (-1)^i \cdot \phi_k(i) \right)$$

$$= y_{t-2}^{\mathrm{STU}} + \sum_{i=1}^{3} M_i^u u_{t+1-i} + \sum_{k=1}^{K} M_k^{\phi+}\sigma_k^{1/4} \left( \bar{U}_{\{t-2:t-1-L\}}\phi_k \right) + \sum_{k=1}^{K} M_k^{\phi-}\sigma_k^{1/4} \left( (\bar{U}_{\{t-2:t-1-L\}} \odot \mathbf{1}^{\pm})\phi_k \right).$$

Using the parameters specified in equation 8 in the above we have that,

$$y_t^{\text{STU}} - y_{t-2}^{\text{STU}} = (CB + D)u_t + CABu_{t-1} - Du_{t-2} + \sum_{l:\alpha_l \geq 0} (\alpha_l + 1)(c_l \otimes b_l) \left(\bar{U}_{\{t-2:t-1-L\}}\right) \left(\underbrace{\sum_{k=1}^{K} (\mu(\alpha_l)^\top \phi_k)\phi_k}_{=\tilde{\mu}(\alpha)}\right)$$

$$+ \sum_{l:\alpha_l < 0} (|\alpha_l| + 1)(c_l \otimes b_l) \left(\bar{U}_{\{t-2:t-1-L\}} \odot \mathbf{1}^{\pm}\right) \left(\underbrace{\sum_{k=1}^{K} (\mu(|\alpha_l|)^\top \phi_k)\phi_k}_{=\tilde{\mu}(|\alpha_l|)}\right)$$

Combining the above display with equation 7, we get that

$$y_t^{\text{LDS}} - y_t^{\text{STU}} = y_{t-2}^{\text{LDS}} - y_{t-2}^{\text{STU}} + \sum_{l:\alpha_l \geq 0} (\alpha_l + 1)(c_l \otimes b_l) \left(\bar{U}_{\{t-2:t-1-L\}}\right) (\mu(\alpha) - \tilde{\mu}(\alpha))$$

$$+ \sum_{l:\alpha_l < 0} (|\alpha_l| + 1)(c_l \otimes b_l) \left(\bar{U}_{\{t-2:t-1-L\}} \odot \mathbf{1}^{\pm}\right) (\mu(|\alpha_l|) - \tilde{\mu}(|\alpha_l|)) \qquad (9)$$

Let $\|B\|_{\text{col}} = \max_l \|b_l\|$, $\|C\|_{\text{col}} = \max_l \|c_l\|$ be the maximum column norms of $B$ and $C$ respectively. Therefore we have that for all $l$, the spectral norm of the matrix $c_l \otimes b_l$ is bounded as $\|B\|_{\text{col}} \cdot \|C\|_{\text{col}}$. Further note that every column of $\bar{U}$ is an input $u_t$ for some time $t$. Further we have assumed that $\|u_t\| \leq a$ for all $t$. Therefore we have that the frobenius norm (and thus spectral norm) of $U_{t-2:t-1-L}$ is bounded as

$$\|\bar{U}_{t-2:t-1-L}\| \leq \|\bar{U}_{t-2:t-1-L}\|_F \leq \sqrt{L} \cdot a.$$

Putting the above together we get that for all $l$,

$$\|(\alpha_l + 1)(c_l \otimes b_l)\left(\bar{U}_{t-2:t-1-L}\right)\| \leq |(\alpha_l + 1)|\|(c_l \otimes b_l)\|\|\left(\bar{U}_{t-2:t-1-L}\right)\| \leq 2 \cdot \|B\|_{\text{col}} \cdot \|C\|_{\text{col}} \cdot \sqrt{L} \cdot a.$$

Therefore we have (using C.3 that,

$$\| \sum_{l:\alpha_l \geq 0} (\alpha_l + 1)(c_l \otimes b_l) \left(\bar{U}_{t-2:t-1-L}\right) (\mu(\alpha) - \tilde{\mu}(\alpha)) \|$$

$$\leq \sum_{l:\alpha_l \geq 0} \|(\alpha_l + 1)(c_l \otimes b_l) \left(\bar{U}_{t-2:t-1-L}\right)\| \cdot \| (\mu(\alpha) - \tilde{\mu}(\alpha)) \|$$

$$\leq 5 \cdot \|B\|_{\text{col}} \cdot \|C\|_{\text{col}} \cdot L^{1.5} \cdot a \cdot \sqrt{\sum_{i=K+1}^{L} \sigma_i}.$$

Similarly we have that

$$\| \sum_{l:\alpha_l < 0} (|\alpha_l| + 1)(c_l \otimes b_l) \left(\bar{U}_{\{t-2:t-1-L\}} \odot \mathbf{1}^{\pm}\right) (\mu(|\alpha_l|) - \tilde{\mu}(|\alpha_l|)) \| \leq 5 \cdot \|B\|_{\text{col}} \cdot \|C\|_{\text{col}} \cdot L^{1.5} \cdot a \cdot \sqrt{\sum_{i=K+1}^{L} \sigma_i}.$$

Plugging the above into equation 9, we get that

$$\|y_t^{\text{LDS}} - y_t^{\text{STU}}\| \leq \|y_{t-2}^{\text{LDS}} - y_{t-2}^{\text{STU}}\| + 10 \cdot \|B\|_{\text{col}} \cdot \|C\|_{\text{col}} \cdot L^{1.5} \cdot a \cdot \sqrt{\sum_{i=K+1}^{L} \sigma_i}$$

Applying the above equation recursively and Lemma C.4 we get that for any $K \geq \log(L)$,

$$\|y_t^{\text{LDS}} - y_t^{\text{STU}}\| \leq 5 \cdot \|B\|_{\text{col}} \cdot \|C\|_{\text{col}} \cdot L^{2.5} \cdot a \cdot \sqrt{\sum_{i=K+1}^{L} \sigma_i} \leq c \cdot \|B\|_{\text{col}} \cdot \|C\|_{\text{col}} \cdot L^3 \cdot a \cdot e^{\left(-\frac{\pi^2}{4} \cdot \frac{K}{\log(L)}\right)}.$$

where $c = 5\Gamma \leq 2 \times 10^6$ is an absolute constant. This finishes the proof of the theorem.

### C.1 Proofs of Lemmas

*Proof of Lemma C.1.* The lemma follows from the following simple calculations.

$$\begin{aligned}
Z(i,j) = \int_0^1 (\alpha - 1)^2 \alpha^{i+j-2} d\alpha &= \int_0^1 \left(\alpha^{i+j} + \alpha^{i+j-2} - 2\alpha^{i+j-1}\right) d\alpha \\
&= \frac{1}{(i+j+1)} + \frac{1}{(i+j-1)} - \frac{2}{(i+j)} \\
&= \frac{2}{(i+j)^3 - (i+j)}
\end{aligned}$$

$\square$

Lemma C.3 is immediate from the second part of Lemma C.2. We show Lemma C.2 below.

*Proof of Lemma C.2.* By definition $\mu(\alpha) = 0$ for $\alpha \in \{0, 1\}$. Otherwise we have that for all $\alpha \in (0, 1)$,

$$|\mu(\alpha)|^2 = \sum_{i=1}^{T} (\alpha - 1)^2 \alpha^{2i-2} \leq \frac{(\alpha-1)^2}{(1-\alpha^2)} \leq \frac{1-\alpha}{1+\alpha} \leq 1 - \alpha$$

To prove the second part we consider drawing $\alpha$ from the uniform distribution between $[0, 1]$. We get that

$$E[(\mu(\alpha)^\top v)^2] = v^\top Z v$$

We now show that the worst case value is not significantly larger than the expectation. To this end we consider the function $f(\alpha) = (\mu(\alpha)^\top v)^2$ and we show that this is a 6-Lipschitz function. To this end consider the following,

$$\begin{aligned}
\left\|\frac{\partial \mu(\alpha)}{\partial \alpha}\right\|_2^2 &= \sum_{i=0}^{T-1} \left\{\left|\frac{\partial}{\partial \alpha}(1-\alpha)\alpha^i\right|^2\right\} \\
&= \sum_{i=0}^{T-1} \left((1-\alpha)i\alpha^{i-1} - \alpha^i\right)^2 \\
&\leq 2(1-\alpha)^2 \sum_{i=1}^{T-1} i^2 \alpha^{2(i-1)} + 2 \sum_{i=0}^{T-1} \alpha^{2i} & (a+b)^2 \leq 2(a^2 + b^2) \\
&\leq 2(1-\alpha)^2 \left(\frac{1}{(1-\alpha^2)^2} + \frac{2\alpha^2}{(1-\alpha^2)^3}\right) + \frac{2}{1-\alpha^2} & \sum_{i=1}^{\infty} i^2 \beta^{i-1} = \frac{1}{(1-\beta)^2} + \frac{2\beta}{(1-\beta)^3} \\
&= \frac{2}{(1+\alpha)^2} + \frac{4\alpha^2}{(1-\alpha^2)(1+\alpha)^2} + \frac{2}{1-\alpha^2}.
\end{aligned}$$

Therefore we have that for all $\alpha \in [0,1]$,

$$\frac{\partial f(\alpha)}{\partial \alpha} = 2(\mu(\alpha)^\top v)\left(\frac{\partial \mu(\alpha)^\top}{\partial \alpha} v\right) \le 2\|\mu(\alpha)\|\|v\|^2\|\frac{\partial \mu(\alpha)}{\partial \alpha}\|$$

$$\le 2\sqrt{(1-\alpha) * \left(\frac{2}{(1+\alpha)^2} + \frac{4\alpha^2}{(1-\alpha^2)(1+\alpha)^2} + \frac{2}{1-\alpha^2}\right)}$$

$$\le 2\sqrt{\left(\frac{2(1-\alpha)}{(1+\alpha)^2} + \frac{4\alpha^2}{(1+\alpha)^3} + \frac{2}{1+\alpha}\right)} \le 6.$$

Now for the positive function $f(\alpha)$ which is 6-Lipschitz on $[0,1]$ let the maximum value be $R$. It can be seen the lowest expected value of $f(\alpha)$ over the uniform distribution over $[0,1]$, one can achieve is $R^2/2 * 6$ and therefore we have that

$$R^2/12 \le v^\top Z v \Rightarrow R \le \sqrt{12 v^\top H v},$$

which finishes the proof.

$\square$

# D  Alternative Representation for capturing negative eigenvalues

In this section we setup an alternative version of STU wherein a different Hankel matrix is used but one can get a similar result. As before a single layer of STU (depicted in figure 2) is parameterized by a number $K$, denoting the number of eigenfactors and matrices $M_1^\phi \ldots M_K^\phi \in \mathbb{R}^{d_{\text{out}} \times d_{\text{in}}}$, and $M_1^u, M_2^u, M_3^u \in \mathbb{R}^{d_{\text{out}} \times d_{\text{in}}}$. The matrices form the *params* of the layer. We use a different Hankel matrix $Z_L \in \mathbb{R}^{L \times L}$ whose entries are given by

$$Z_L[i,j] \triangleq ((-1)^{i+j-2} + 1) \cdot \frac{8}{(i+j+3)(i+j-1)(i+j+1)}. \tag{10}$$

and let $\{(\sigma_j \in \mathbb{R}, \phi_j \in \mathbb{R}^T)\}_{j=1}^T$ be the eigenvalue-eigenvector pairs of $Z_L$ ordered to satisfy $\sigma_1 \ge \sigma_2 \ldots \sigma_d$.

Given an input sequence $\{u_1 \ldots u_L\} \in \mathbb{R}^{d_{\text{in}}}$, as before we first featurize the input sequence by *projecting* the input sequence till time $t$ on *fixed* filters $\phi_k$. The main difference is that we do not need to create a negative featurization now. We define

$$U_{t,k} = \sum_{i=0}^{t-1} u_{t-i} \cdot \phi_k(i).$$

Note that for every $k$, the sequence of features $X_{1:T,k}$ can be computed efficiently via convolution. The output sequence $\{y_1 \cdots y_T\}$ is then given by

$$\hat{y}_t = \underbrace{\hat{y}_{t-2} + \sum_{i=1}^{3} M_i^u u_{t+1-i}}_{\text{Auto}-\text{regressive Component}} + \underbrace{\sum_{k=1}^{K} M_k^\phi \sigma_k^{1/4} X_{t-2,k}}_{\text{Spectral Component}}. \tag{11}$$

We prove the following representation theorem which shows that the above class approximately contains any marginally-stable LDS with symmetric $A$.

**Theorem D.1.** *Given any $A, B, C, D$ such that $A$ is a symmetric matrix with $\|A\| \le 1$ and given any numbers $K \in \mathbb{I}^+, a \in \mathbb{R}^+$, there exists matrices $M_1^u, M_2^u, M_3^u, M_1^\phi \ldots M_K^\phi \in \mathbb{R}^{d_{\text{out}} \times d_{\text{in}}}$ for all $L$ and all sequences $u_{1:L}$ satisfying $\|u_t\| \le a$ for all $t \in [L]$ the following holds. Let $y_{1:L}^{\text{LDS}}$ be the sequence generated by execution of the LDS given by $A, B, C, D$ (via equation 1) and $y_{1:L}^{\text{SF}}$ be the sequence generated by Spectral Filtering (via equation 11) using the matrices $M_1^u, M_2^u, M_3^u, M_1^{\phi+} \ldots M_K^{\phi+}, M_1^{\phi-} \ldots M_K^{\phi-}$. Then for all $t \in [T]$, we have that*

$$\|y_t^{\text{LDS}} - y_t^{\text{SF}}\|^2 \le c \cdot \|B\|_{col} \cdot \|C\|_{col} \cdot L^3 \cdot a \cdot e^{-\left(\frac{\pi^2}{4} \cdot \frac{K}{\log(L)}\right)}$$

*where $c \le 10^6$ is a universal constant and $\|B\|_{col}, \|C\|_{col}$ are the maximum column norm of the matrices $B$ and $C$ respectively.*

In the following we prove the above theorem.

### D.1 Proof of Theorem D.1

Without loss of generality we assume that $A$ is a real-diagonal matrix. Before continuing with the proof, we will provide some requisite definitions and lemmas. Define the following vector for any $\alpha$, $\mu(\alpha) \in \mathbb{R}^T$, with $\mu(\alpha)(i) = (\alpha^2 - 1)\alpha^{i-1}$. Further define the Hankel matrix $H$ as

$$Z \triangleq \int_{-1}^{1} \mu(\alpha)\mu(\alpha)^\top d\alpha$$

As the following lemma shows the Hankel matrix $Z$ above is the same Hankel matrix $Z_L$ defined in the definition of STU equation 10.

**Lemma D.2.** $Z$ is a Hankel matrix with entries given as

$$Z(i, j) = ((-1)^{i+j-2} + 1) \cdot \frac{8}{(i+j+3)(i+j-1)(i+j+1)}$$

*Proof.* Consider the following simple computations

$$\begin{aligned}
H(i, j) &= \int_{-1}^{1} (\alpha^2 - 1)^2 \alpha^{i+j-2} d\alpha \\
&= \int_{-1}^{0} (\alpha^2 - 1)^2 \alpha^{i+j-2} d\alpha + \int_{0}^{1} (\alpha^2 - 1)^2 \alpha^{i+j-2} d\alpha \\
&= \int_{-1}^{0} (|\alpha|^2 - 1)^2 (-1)^{i+j-2} |\alpha|^{i+j-2} d\alpha + \int_{0}^{1} (\alpha^2 - 1)^2 \alpha^{i+j-2} d\alpha \\
&= \int_{0}^{1} (\alpha^2 - 1)^2 (-1)^{i+j-2} \alpha^{i+j-2} d\alpha + \int_{0}^{1} (\alpha^2 - 1)^2 \alpha^{i+j-2} d\alpha \\
&= ((-1)^{i+j-2} + 1) \int_{0}^{1} (\alpha^2 - 1)^2 \alpha^{i+j-2} d\alpha \\
&= ((-1)^{i+j-2} + 1) \cdot \frac{8}{(i+j+3)(i+j-1)(i+j+1)}
\end{aligned}$$

$\square$

**Lemma D.3.** *We have that the following statements hold regarding $\mu(\alpha)$ for any $\alpha \in [-1, 1]$,*

- $|\mu(\alpha)|^2 \le 1$

- *For any $\alpha \in [-1, 1]$ and any unit vector $v$ we have that*

$$(\mu(\alpha)^\top v)^2 \le 6(v^\top Z v)$$

*Proof.* By definition $\mu(\alpha) = 0$ for $\alpha \in \{-1, 1\}$. Otherwise we have that for all $\alpha \in (-1, 1)$,

$$|\mu(\alpha)|^2 = \sum_{i=1}^{T} (\alpha^2 - 1)^2 \alpha^{2i-2} \le \frac{(\alpha^2 - 1)^2}{(1 - \alpha^2)} = 1 - \alpha^2 \le 1.$$

To prove the second part we consider drawing $\alpha$ from the uniform distribution between $[-1, 1]$. We get that

$$E[(\mu(\alpha)^\top v)^2] = \frac{v^\top Z v}{2}$$

We now show that the worst case value is not significantly larger than the expectation. To this end we consider the function $f(\alpha) = (\mu(\alpha)^\top v)^2$ and we show that this is a 6-Lipschitz function. To this end consider the following,

$$
\begin{aligned}
\left\|\frac{\partial\mu(\alpha)}{\partial\alpha}\right\|_2^2 &= \sum_{i=0}^{T-1}\left\{\left|\frac{\partial}{\partial\alpha}(1-\alpha^2)\alpha^i\right|^2\right\} \\
&= \sum_{i=0}^{T-1}\left((1-\alpha^2)i\alpha^{i-1} - 2\alpha^{i+1}\right)^2 \\
&\leq 2(1-\alpha^2)^2\sum_{i=1}^{T-1}i^2\alpha^{2(i-1)} + 4\sum_{i=0}^{T-1}\alpha^{2i+2} && (a+b)^2 \leq 2(a^2+b^2) \\
&\leq 2(1-\alpha^2)^2\left(\frac{1}{(1-\alpha^2)^2} + \frac{2\alpha^2}{(1-\alpha^2)^3}\right) + \frac{4\alpha^2}{1-\alpha^2} && \sum_{i=1}^{\infty}i^2\beta^{i-1} = \frac{1}{(1-\beta)^2} + \frac{2\beta}{(1-\beta)^3} \\
&= 2 + \frac{8\alpha^2}{(1-\alpha^2)}.
\end{aligned}
$$

Therefore we have that for all $\alpha \in [-1, 1]$,

$$
\begin{aligned}
\frac{\partial f(\alpha)}{\partial\alpha} &= 2(\mu(\alpha)^\top v)\left(\frac{\partial\mu(\alpha)^\top}{\partial\alpha}v\right) \leq 2\|\mu(\alpha)\|\|v\|^2\|\frac{\partial\mu(\alpha)}{\partial\alpha}\| \\
&\leq 2\sqrt{(1-\alpha^2)*\left(2 + \frac{8\alpha^2}{(1-\alpha^2)}\right)} \\
&\leq 2\sqrt{2+6\alpha^2} \leq 6.
\end{aligned}
$$

Now for the positive function $f(\alpha)$ which is 6-Lipschitz on $[-1, 1]$ let the maximum value be $R$. It can be seen the lowest expected value of $f(\alpha)$ over the uniform distribution over $[0, 1]$, one can achieve is $R^2/2*6$ and therefore we have that

$$
R^2/12 \leq \frac{v^\top Z v}{2} \Rightarrow R \leq \sqrt{6v^\top Z v},
$$

which finishes the proof. $\qquad\square$

A direct consequence of the above lemma is the following.

**Lemma D.4.** *For any $\alpha \in [0, 1]$, let $\tilde{\mu}(\alpha)$ be the projection of $\mu(\alpha)$ on the subspace spanned by top $k$ eigenvectors of $Z$, then we have that*

$$
\|\mu(\alpha) - \tilde{\mu}(\alpha)\|^2 \leq 6\sum_{i=k+1}^{L}\sigma_i
$$

Finally the following lemma with a proof similar to C.3 shows that the spectrum of the matrix $Z$ decays exponentially.

**Lemma D.5.** *Let $\sigma_j$ be the top $j^{th}$ eigenvalue of $Z$. Then we have that*

$$
\sigma_j \leq \Gamma c^{-j/\log(L)}
$$

*where $c = e^{\pi^2/4} \sim 11.79$ and $\Gamma = 235200$ is an absolute constant.*

We now move towards proving Theorem D.1. Consider the following calculation for the LDS sequence $y_t^{\text{LDS}}$

$$
y_t^{\text{LDS}} = \sum_{i=0}^{T}CA^iBu_{t-i} + Du_t,
$$

and therefore we have that

$$y_t^{\text{LDS}} - y_{t-2}^{\text{LDS}} = (CB + D)u_t + CABu_{t-1} - Du_{t-2} + \underbrace{\sum_{i=0}^{T} C(A^{i+2} - A^i)Bu_{t-2-i}}_{\text{Term of Interest}}$$

For any $t_1 \geq t_2$ we define the matrix $\bar{U}_{t_1:t_2} \in \mathbb{R}^{d_{\text{out}} \times t_1 - t_2 + 1}$ whose $i^{th}$ column is the input vector $u_{t_1-i+1}$. We allow $t_2$ to be negative and by convention assume $u_t = 0$ for any $t \leq 0$. Denote the diagonal entries of $A$ by $\{\alpha_l\}_{l=1}^{d_h}$, i.e. $\alpha_l = A(l, l)$. The term of interest above can then be written as

$$\sum_{i=0}^{L} C(A^{i+2} - A^i)Bu_{t-2-i} = \sum_{l=1}^{d_h}(c_l \otimes b_l) \left( \sum_{i=0}^{L}(\alpha_l^{i+2} - \alpha_l^i)u_{t-2-i} \right)$$

$$= \sum_{l=1}^{d_h}(c_l \otimes b_l) \left( \sum_{i=0}^{L}(\alpha_l^2 - 1)\alpha_l^i u_{t-2-i} \right)$$

$$= \sum_{l=1}^{d_h}(c_l \otimes b_l) \left( \bar{U}_{\{t-2:t-1-L\}}\mu(\alpha) \right).$$

Therefore we get that

$$y_t^{\text{LDS}} - y_{t-2}^{\text{LDS}} = (CB + D)u_t + CABu_{t-1} - Du_{t-2} + \sum_{l=1}^{d_h}(c_l \otimes b_l) \left( \bar{U}_{\{t-2:t-1-L\}}\mu(\alpha) \right).$$

Recall that we defined the sequence $\{\sigma_k, \phi_k\}_{k=1}^{L}$ to be the eigenvalue and eigenvector pairs for the Hankel matrix $Z$. For any $\alpha$ we define the projection of $\mu(\alpha)$ on the top $k$ eigenvectors as $\tilde{\mu}(\alpha)$, i.e. $\tilde{\mu}(\alpha) = \sum_{k=1}^{K}(\mu(\alpha_l)^\top \phi_k)\phi_k$. Further define STU parameters as follows

$$M_1^u = CB + D, M_2^u = CAB, M_3^u = -D$$

$$M_k^\phi = \sum_l (\mu(\alpha_l)^\top \phi_k)\sigma_k^{-1/4}(c_l \otimes b_l)$$

The definition of STU prediction (using the above parameters) implies that the predicted sequence satisfies

$$y_t^{\text{STU}} - y_{t-2}^{\text{STU}} = (CB + D)u_t + CABu_{t-1} - Du_{t-2} + \sum_l(c_l \otimes b_l) \left( \bar{U}_{\{t-2:t-1-L\}} \right) \left( \underbrace{\sum_{k=1}^{K}(\mu(\alpha_l)^\top \phi_k)\phi_k}_{=\tilde{\mu}(\alpha)} \right).$$

Combining the above displays we get that

$$y_t^{\text{LDS}} - y_t^{\text{STU}} = y_{t-2}^{\text{LDS}} - y_{t-2}^{\text{STU}} + \sum_l(c_l \otimes b_l) \left( \bar{U}_{\{t-2:t-1-L\}} \right) (\mu(\alpha) - \tilde{\mu}(\alpha)).$$

Using a similar derivation as in the proof of Theorem 3.1 we get that

$$\|y_t^{\text{LDS}} - y_t^{\text{STU}}\| \leq \|y_{t-2}^{\text{LDS}} - y_{t-2}^{\text{STU}}\| + 10 \cdot \|B\|_{\text{col}} \cdot \|C\|_{\text{col}} \cdot L^{1.5} \cdot a \cdot \sqrt{\sum_{i=K+1}^{L} \sigma_i}$$

Applying the above equation recursively and Lemma D.5 we get that for any $K \geq \log(L)$

$$\|y_t^{\text{LDS}} - y_t^{\text{STU}}\| \leq 5 \cdot \|B\|_{\text{col}} \cdot \|C\|_{\text{col}} \cdot L^{2.5} \cdot a \cdot \sqrt{\sum_{i=K+1}^{L} \sigma_i} \leq c \cdot \|B\|_{\text{col}} \cdot \|C\|_{\text{col}} \cdot L^3 \cdot a \cdot e^{\left(-\frac{\pi^2}{4} \cdot \frac{K}{\log(L)}\right)}.$$

where $c = 2.5 \times \Gamma \leq 10^6$ is an absolute constant. This finishes the proof of the theorem.

# E   Experiment Details

## E.1   Synthetic Experiments with a marginally-stable LDS

The random system we generated for the experiments displayed in Figure 3a is as follows -

$$
A = \begin{bmatrix} -0.9999 & 0. & 0. & 0. \\ 0. & 0.9999 & 0. & 0. \\ 0. & 0. & -0.9999 & 0. \\ 0. & 0. & 0. & 0.9999 \end{bmatrix}, \quad
B = \begin{bmatrix} 0.36858183 & -0.34219486 & 0.1407376 \\ 0.18933886 & -0.1243964 & 0.21866894 \\ 0.14593862 & -0.5791096 & -0.06816235 \\ -0.3095346 & -0.21441863 & 0.08696061 \end{bmatrix}
$$

$$
C = \begin{bmatrix} 0.5528727 & -0.51329225 & 0.21110639 & 0.2840083 \\ -0.18659459 & 0.3280034 & 0.21890792 & -0.8686644 \\ -0.10224352 & -0.46430188 & -0.32162794 & 0.1304409 \end{bmatrix}, \quad
D = \begin{bmatrix} 1.5905786 & 0. & 0. \\ 0. & -0.45901108 & 0. \\ 0. & 0. & 0.3238576 \end{bmatrix}
$$

**Hyperparameters for STU:**   We only tuned the learning rate in the set $([5e-2, 1e-1, 5e-1, 1, 5, 10])$ for vanilla STU and used $K = 25$.

**Hyperparameters for LRU:**

- **Model Hyperparameters** Orvieto et al. (2023) provide a few recommendations for the LRU model. We tested exhaustively over the following hyperparameter choices:

    - Stable Exp-parameterization: We searched over [True, False]

    - Logarithmic Representation of Recurrent Parameters: We searched over [True, False]

    - $\gamma$-Normalization: We searched over [True, False]

    - Ring Initialization:   We searched over `min_rad`$\in$ $\{0.0, 0.9, 0.99, 0.999\}$ and `max_rad`$\in$ $\{0.9, 0.99, 0.999, 1.0\}$.

    - Setting the `max_init_phase`$\in \{1.57, 3.14, 6.28\}$

    We found the Stable Exp-parameterization, Logarithmic Representation of Recurrent Parameters and $\gamma$-normalization to be essential for training in this problem. We did not observe any particular benefit of Ring Initialization or reducing the phase at initialization and we set them to defaults eventually. We provide the learning curves over our search space in Figure 6.

- **Optimization Hyperparameters** Given the comparatively higher sample complexity of the LRU model we employed standard deep-learning optimization tricks like tuning weight-decay as well as applying a cosine learning rate schedule with warmup. These optimization tricks did not lead to gains over standard training with Adam and a fixed learning rate in this problem. We tuned the learning rate in the set $([5e-2, 1e-1, 5e-1, 1, 5, 10])$.

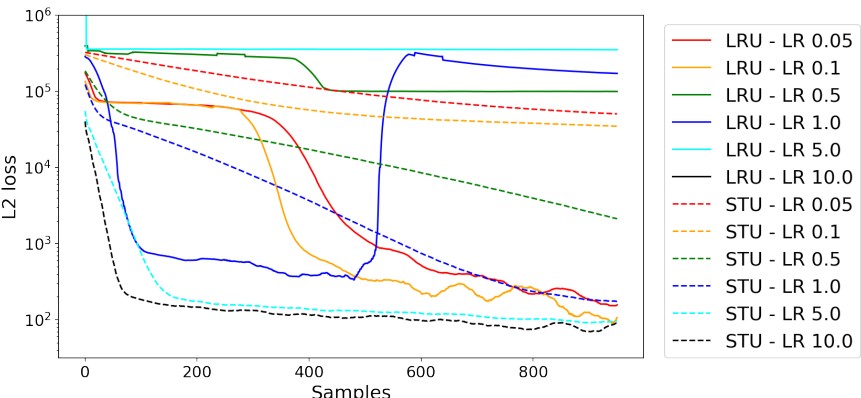

Figure 5: (Smoothed) Learning curves for learning a marginally stable LDS for a single STU layer (dashed) vs a single LRU layer (solid). Different colors represent different learning rates highlighting that the training becomes unstable for LRUs quickly as LR increases while the STU trains at much higher learning rates. Curiously at stable LRs we observe that LRUs show a platea-ing of learning for a large fraction of the training time.

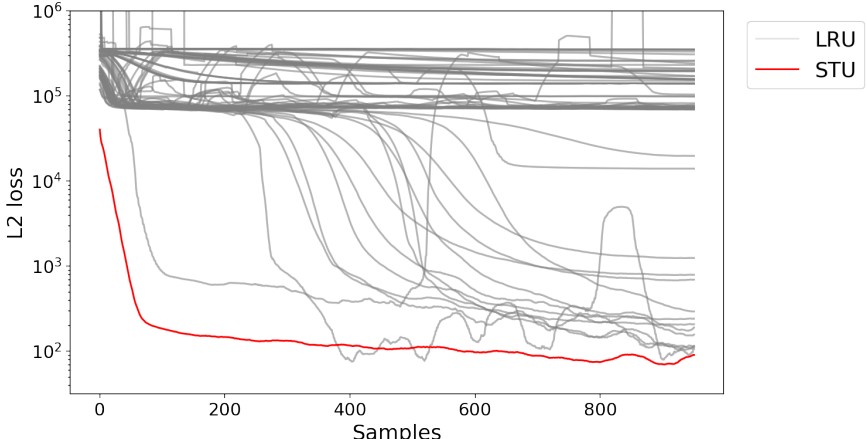

Figure 6: LRU Hparam search vs STU. All the gray curves represent the hyperparameters for LRU we tried. The STU curve is the best taken from Figure 5. For LRU we searched over choices of enabling stable exp-parameterization, gamma-normalization, ring-initialization, phase-initialization, learning rate, weight decay and constant vs warmup+cosine decay lr schedule.

### E.2 Experimental setup for LRA experiments

Our training setup closely follows the experimental setup used by Orvieto et al. (2023). We use the same batch sizes and training horizons for all the tasks as employed by Orvieto et al. (2023).

**Hyperparameter tuning** For all of our experiments on the LRA benchmark for both the vanilla STU model and the auto-regressive AR-STU model we searched the learning rate in the set $\{1e-4, 3e-4, 5e-4, 1e-3, 2.5e-3, 5e-3\}$ and tune the weight decay in the set $\{1e-3, 1e-2, 1e-1, 5e-1, 1.0\}$. We fix the number of filters $K$ to be 24. We use Adam as the training algorithm with other optimization hyperparameters set to their default values. We use the same learning rate schedule as Orvieto et al. (2023), i.e. 10% warmup followed by cosine decay to 0. For the AR-STU model we searched over two values of $k_y \in \{2, 32\}$. In Table 2 we present a comparison of vanilla STU with AR-STU with $k_y = 2$ and AR-STU with $k_y = 32$. We find that both vanilla STU and AR-STU $k_y = 2$ reach comparable accuracy which is better than the baselines S4

| | CIFAR | ListOps | Text | Retrieval | Pathfinder | PathX |
|---|---|---|---|---|---|---|
| S4 Gu et al. (2021a) | 88.65 | 59.60 | 86.82 | 90.90 | 94.20 | 96.35 |
| LRU Orvieto et al. (2023) | 89 | 60.2 | 89.4 | 89.9 | 95.1 | 94.2 |
| STU | 83.73 | 61.04 | **90.48** | 90.40 | 91.70 | 89.71 |
| AR-STU ($k_y = 2$) | 86.56 | **61.14** | **90.47** | 90.52 | 93.85 | 90.49 |
| AR-STU ($k_y = 32$) | **91.34** | 57.66 | 88.51 | 87.39 | **95.45** | 93.24 |

Table 2: Comparison of the STU model against various proposed SSM models on the LRA benchmark: Bold values indicate the best for that task. We find that STU is competitive across all the workloads without the need for carefully designed initializations, discretizations or normalizations. We report the median over 5 trials for our experiments.

and LRU on non-image datasets. On image datasets we found $k_y = 32$ to be helpful in getting better test accuracies.

**Initialization** For the STU model we initialized **all** the $M$ matrices at 0.

Finally while training the AR-STU model as employed by the training setup of Orvieto et al. (2023) and previous SSM implementations, we found that using a smaller value of LR specifically for $M^y$ matrices to be useful. We decreased the value of LR by a factor 0.1 or 0.05 and searched over this parameter.

## F  Power of Auto-regression: Dimension-dependent representation for LDS

In this section we give a short proof that any partially-observed LDS can be perfectly predicted via a linear predictor acting over at most $d$ of its past inputs and outputs where $d$ is the hidden-state dimensionality (i.e. $A \in \mathbb{R}^{d \times d}$). In particular

**Theorem F.1.** *Given an LDS parameterized by $A \in \mathbb{R}^{d \times d}, B, C, D$, there exist coefficients $\alpha_{1:d}$ and matrices $\Gamma_{0:d}$ such that given any input sequence $u_{1:L}$, the output sequence $y_{1:L}$ generated by the action of the LDS on the input satisfies for all $t$*

$$y_t = \sum_{i=1}^{d} \alpha_i y_{t-i} + \sum_{i=0}^{d} \Gamma_i u_{t-i}$$

*Proof.* By unrolling the LDS we have that $y_t = \sum_{i=0}^{t} CA^i B u_{t-i} + D u_t.$. By the Cayley Hamilton theorem, the matrix $A$ has a characteristic polynomial $p$ of degree $d$, namely there exists $d$ numbers $c_{1:d}$ such that

$$p(z) = \sum_{i=0}^{d} c_i z^i$$

satisfies $p(A) = 0$. Without loss of generality we can assume the constant term in the polynomial is 1. We can now consider the series for $y_t, y_{t-1}, \ldots$ as

$$
\begin{array}{rclcllc}
y_t - D u_t & = & CB u_t & + & CAB u_{t-1} & + & \ldots & CA^t B u_1 \\
y_{t-1} - D u_{t-1} & = & 0 & + & CB u_{t-1} & + & \ldots & CA^{t-1} B u_1 \\
\vdots & & & & & & & \\
y_{t-d} - D u_{t-d} & = & 0 & + & 0 & + & \ldots & CA^{t-d} B u_1
\end{array}
$$

Now, if we take the combination of the above rows according to the coefficients of the characteristic polynomial, we get that

$$\sum_{i=0}^{d} c_i y_{t-i} = \sum_{j=0}^{t} R_j + \sum_{i=0}^{d} D u_{t-i} \tag{12}$$

where $R_j$ is the appropriate sum along the $j'th$ column of the matrix above. For all $j > d$, this amounts to an expression of the form:

$$j > d \quad \Rightarrow \quad R_j = \sum_{i=0}^{d} c_i C A^i \cdot A^{t-j} B u_{t-j} = C(\sum_{i=0}^{d} c_i A^i) \cdot A^{t-j} B u_{t-j} = C \cdot p(A) \cdot A^{t-j} B u_{t-j} = 0.$$

Since all but the first $d$ columns are zero, rearranging equation 12 and collecting terms, we get that there exists coefficients $\alpha_{1:d}$ and matrices $\Gamma_{0:d}$ such that

$$y_t = \sum_{i=1}^{d} \alpha_i y_{t-i} + \sum_{j=0}^{d} \Gamma_j u_{t-j}.$$

$\square$

