# OpenReview forum: "Spectral State Space Models"
_TMLR — Rejected by TMLR_

### Review · Reviewer_kdkQ · 2024-10-04

**Summary Of Contributions:**

This paper extends previous work on spectral filtering for linear dynamical systems (LDS) into a stacked deep neural network architecture. Theoretically, it extends the approximation of spectral filtering for LDS by relaxing the previous positive semi-definite constraint on the state transition matrix A for LDS to include all symmetric matrices. This is achieved by incorporating a negative part in the spectral component and also adding a few autoregressive components, which finally yields a unit named as Spectral Transform Unit (STU). Empirically, the paper verifies that a single STU layer can approximate a marginally stable LDS synthetic data with faster and more stable convergence than a single Linear Recurrent Unit (LRU) layer. Additionally, the proposed stacked spectral STU unit consistently outperforms S4 and LRU in experiments on the long-range arena benchmark for visual and textual modeling in sequence to sequence tasks.

I have thoroughly read the main text, including the formulations, theorems, and experiments. However, I acknowledge that I have not examined the details in the proofs and other sections in the appendix.

I would recommend acceptance.

**Audience:**

Yes

**Broader Impact Concerns:**

Not applicable.

**Claims And Evidence:**

Yes

**Requested Changes:**

Please kindly refer to the weakness section.

**Strengths And Weaknesses:**

Strength 1: The paper leverages the strength of the spectral filtering algorithm, which does not depend on either the spectrum of the linear dynamics or the dimensionality of the problem. Additionally, its approximation to LDS requires a number of filters that scale logarithmically with respect to the length of the discrete sequence. This allows it to model very long dependencies in sequence-to-sequence tasks.

Strength 2: The spectral filters are fixed and do not require learning.

Strength 3: The paper leverages the strength of spectral filtering to more easily learn long-term memory and dependencies in linear dynamical systems.

Strength 4: The proposed STU does not require specific initializations, discretizations, or normalizations, which are necessary for existing state-space models (SSMs) for modeling.

Strength 5: The paper removes the positive semi-definite (psd) constraints required in previous theoretical bounds, extending them to general symmetric real matrices. This is achieved by including autoregressive components and negative spectral components.

Strength 6: The one-layer STU model performs well on synthetic data for modeling LDS, and the stacked model outperforms the previous S4 model for modeling real-world data on the long-range arena benchmark.

Strength 7: The paper is generally well-written, well-structured, and easy to read.



Weakness 1: Although the approximation constraint of spectral filtering for linear dynamical systems was relaxed from the previous positive semi-definite state transition matrix (A for LDS) to general symmetric matrices, theoretical bounds for general real matrices have not yet been established.

Weakness 2: The model would benefit from more rigorous testing and benchmarking on a wider range of real-world datasets, and also through direct comparisons against transformer models.

Weakness 3: While much appreciated for sequence-to-sequence modeling, further exploration of the model’s applicability to other tasks would be beneficial.

---

> ### Author Response · Authors · 2024-11-19
> **Thanks for the review**
>
> We thank the reviewer for their positive feedback. We have provided a response to both the points raised by the reviewer in the responses to the other reviewers (as they raised the same points as well). We repeat them here –
>
> * Regarding System Dynamics being symmetric – indeed there can be asymmetric dynamics in the world, but to what extent is the symmetric dynamics a restriction in generative models currently is an open question. In its current state models like Mamba/Griffin and various others, parameterize the dynamics matrices as real diagonal matrices which highlights that symmetric dynamics seem sufficient for generative models currently. We would like to add that there are avenues to naturally extend the method to complex dynamics using the ideas presented in Hazan et al (https://arxiv.org/abs/1802.03981). In a more general sense, we would also like to stress that there are lower bounds for learning general linear dynamical systems w. asymmetric transition matrices which also apply to other SSMs like Mamba/S4. The lower bound necessitates d_hidden auto-regressive components and cannot be improved. We will add a discussion in the paper regarding this point.
>
> - Regarding further experiments – Indeed it is an active area of research for us. We would like to specifically highlight the following
>   - Regarding comparisons with Transformers – an independent follow-up work has established the efficacy of Hybrid-Transformer-SSM models against GPT like models which can be found here – Flash STU: Fast Spectral Transform Units - Liu et al. (https://arxiv.org/pdf/2409.10489). Indeed the paper finds that an STU-Transformer hybrid improves upon vanilla transformers significantly.
>   - The same paper as the one linked above performs a comparison of STU (+ other models) on special tasks like Selective Copy and Induction Heads etc. Specifically regarding Selective Copy (Figure 20) it shows that both STU and STU-Transformer hybrid models can solve this task (owing to the sandwiched non-linearity in between layers)
>   - Regarding additional results on diverse real-world datasets, we would like to add that indeed further experiments are an area of active research for us.

---

> > ### Comment · Reviewer_kdkQ · 2024-11-25
> > **Thanks for the Response**
> >
> > I would like to thank the authors for their response to my review.

---

### Review · Reviewer_4PQP · 2024-10-11

**Summary Of Contributions:**

The paper presents a novel neural architecture based on spectral state space models for efficiently modeling sequences with long-range dependencies. The architecture is characterized by its capacity for stable learning without needing complex parameterizations or initializations. This innovation stands out by using theoretically fixed kernels, enhancing computational efficiency, and maintaining expressiveness irrespective of system dimensions or spectral gaps, as opposed to other convolutional models.

This architecture excels at modeling sequential data, particularly for tasks involving long-range dependencies. The key aspects of this contribution include:

1. Spectral State Space Models: The architecture utilizes spectral methods to model sequences, which allows for stable and efficient learning without the need for complex exponential parameterizations, special initializations, or normalizations typically required in other models.
2. Parameter-Free Kernels: Unlike other convolutional models that learn kernels from data, this approach employs fixed, theoretically-grounded kernels. This significantly reduces the complexity associated with training, as there are no parameters to learn for the kernels, enhancing computational efficiency.
3. Expressiveness and Stability: The models are designed to be as expressive as a linear dynamical system (LDS). Their expressiveness does not depend on the spectral gap or system dimensions, which are limitations in other methods. This ensures that the architecture can effectively model data with varying characteristics while maintaining stability.
4. Implementation and Application: The neural architecture has been implemented and tested on synthetic data and real-world benchmarks like the Long Range Arena (LRA). The results demonstrate its ability to efficiently handle sequence modeling tasks, achieving high accuracy on challenging tasks like PathFinder and PathX without requiring additional normalizations or special initializations.

**Audience:**

Yes

**Claims And Evidence:**

Yes

**Requested Changes:**

Please address the weakness above.

**Strengths And Weaknesses:**

S1. The architecture supports stable learning of long-range dependencies in sequences due to its design leveraging spectral state space models. This is a significant advancement in reducing complexities found in other methods that might require complex initializations or parameter tuning.

S2. Unlike many models that learn kernels from data, the spectral state space models use fixed, theoretically-founded kernels. This reduces the parameter burden and enhances the model's efficiency, making it theoretically as expressive as a linear dynamical system (LDS).

S3. The application of this architecture to synthetically generated data and benchmarks like the Long Range Arena demonstrates its ability to handle sequence modeling tasks effectively, thus proving its practical utility and versatility in handling various data types.

W1. Despite its advancements, the paper may present a steep learning curve for readers unfamiliar with spectral methods or state space models. The sophisticated mathematical concepts, like eigenvalue handling and spectral filtering, might limit accessibility for a broader audience.

W2. The implementation assumes knowledge of the eigenvalues being real numbers, which might not always be feasible in real-world applications where system dynamics are not fully understood or are highly complex.

W3. While the architecture has been tested on synthetic data and a specific benchmark, the range of experiments might be limited. Additional results on diverse real-world datasets would strengthen the claims of generalizability and efficiency.

---

> ### Author Response · Authors · 2024-11-19
> **Thanks for the review**
>
> We thank the reviewer for their review and especially appreciating the innovation of the method presented in the paper. We respond to the listed weaknesses below –
>
> * Regarding steep learning curve – We will incorporate this feedback and include appropriate background in the camera ready.
>
> * Regarding System Dynamics being symmetric – indeed asymmetric dynamics are ubiquitous in the physical world, but to what extent is the symmetric dynamics a restriction in generative models currently is an open question. In its current state models like Mamba/Griffin and various others, parameterize the dynamics matrices as real diagonal matrices which highlights that symmetric dynamics seem sufficient for generative models currently. We would like to add that there are avenues to naturally extend the method to complex dynamics using the ideas presented in Hazan et al (https://arxiv.org/abs/1802.03981). In a more general sense, we would also like to stress that there are lower bounds for learning general linear dynamical systems w. asymmetric transition matrices which also apply to other SSMs like Mamba/S4. The lower bound necessitates d_hidden auto-regressive components and cannot be improved. We will add a discussion in the paper regarding this point.
>
> * Regarding additional results on diverse real-world datasets, we would like to add that indeed further experiments are an area of active research for us. We refer the reviewer to the response to the first reviewer for a more detailed discussion of this point.

---

### Review · Reviewer_idwP · 2024-10-31

**Summary Of Contributions:**

The paper introduces a novel approach for handling long-range dependencies in sequence modeling by reformulating state space models (SSMs) via spectral filtering. The authors show that the proposed spectral filtering is a robust representation, theoretically unaffected by the underlying spectral properties of system matrices. The architecture comprises fixed convolutional filters, thus alleviating the need for learned filters and maintaining competitive performance across sequence tasks.

In terms of experiments, the architecture is validated through synthetic dynamical systems and the Long Range Arena benchmark, demonstrating both stability and efficiency without dependency on specialized initializations or regularization.

**Audience:**

Yes

**Broader Impact Concerns:**

I am not sure if this work needs a Broader Impact section. It's a fairly theoretical work introducing a spectral mechanism for State Space Models that can be very generally applied.

**Claims And Evidence:**

Yes

**Requested Changes:**

1. The paper is already quite strong in terms of the theoretical foundations. Although, the empirical evaluations could be expanded by including other long-sequence tasks. For example, "Mamba: Linear-Time Sequence Modeling with Selective State Spaces, Gu et al." includes comparisons on **extremely long sequences, e.g. genomics** (HG38 DNA sequences), or **audio generation** (YouTubeMix and SC09) tasks. I wonder how spectral SSMs could be adapted to these tasks and their respective performance.
2. The authors do mention that Transformer models have been outperformed by SSMs on the LRA tasks. However, it would be interesting to see how spectral SSM compares to the recent long-context transformers on other benchmarks, such as the ones mentioned in the point above.
3. Can spectral SSMs be adapted to or maintain the performance on specialized tasks like "Selective Copying" (which has been used in the Mamba paper)?
4. Also, would it make sense to compare the method to Mamba or "RMKV: Reinventing rnns for the transformer era, Peng et al."?

**Strengths And Weaknesses:**

# Strengths
1. **Theoretical Foundation**: The use of spectral filtering provides a robust, theoretically sound alternative to existing SSMs, particularly advantageous for long-range memory tasks. The independence of model performance from spectral gap or dimensionality enhances stability and broad applicability.
2. The authors show that the number of required filters grow logarithmically in the sequence length L, which highlights the efficiency of the spectral representation. It's also interesting that the spectral filtering process yields a convex problem, allowing for stable and efficient optimization, which is rare in non-convex state space formulations.
3. **Empirical Validation**: Results on synthetic data and the Long Range Arena benchmark corroborate the theoretical benefits, with the proposed model demonstrating high accuracy and efficiency.

# Weaknesses
Weaknesses and suggestions have been listed in the next section.

---

> ### Comment · Reviewer_idwP · 2024-11-16
> **Response requested from the authors**
>
> It would be great if the authors can respond to the weaknesses listed above, so that I can make an informed recommendation.

---

> ### Author Response · Authors · 2024-11-19
> **Thanks for the review**
>
> We thank the reviewer for their review and especially appreciating the theoretical soundness of the method presented in the paper.
>
> * Indeed it makes sense to compare the method to Mamba/RMKV on genomics/audio tasks. At this point we consider this future work, which we are actively working towards. The scope of this paper is to present the underlying ideas and theoretical developments and perform the first round of verifications much the same as the S4/LRU (which were precursors to Mamba/Griffin) performed. Indeed, as stressed in the paper, the ability to learn the PathX task with little modifications bodes well for the ability of STU at longer contexts.
>
> * Regarding comparisons with Transformers – an independent follow-up work has established the efficacy of Hybrid-Transformer-SSM models against GPT like models which can be found here – Flash STU: Fast Spectral Transform Units - Liu et al. (https://arxiv.org/pdf/2409.10489). Indeed the paper finds that an STU-Transformer hybrid improves upon vanilla transformers significantly.
>
> * The same paper as the one linked above performs a comparison of STU (+ other models) on special tasks like Selective Copy and Induction Heads etc. Specifically regarding Selective Copy (Figure 20) it shows that both STU and STU-Transformer hybrid models can solve this task (owing to the sandwiched non-linearity in between layers)
>
> We hope our response answers some of the requested experiments for the author. We reiterate, the scope of the submitted work is to present the main ideas, intuitions, theoretical developments and the first set of experimentation establishing the efficacy of the method.

---

### Decision · Action_Editor_VuB8 · 2025-02-10

**Recommendation:** Reject

**Comment:**

The paper should more clearly spell out its claimed contributions. In addition, if the authors want to keep the current main claim, i.e.,
"We demonstrate that spectral state space models can stably and more efficiently learn on sequence modeling tasks with long range dependencies without the need for exponential parameterizations, particular initializations and normalizations.", the empirical assessment of the proposed approach should be improved in order  to have a solid experimental support to backup it. Especially the claim that fixed convolutional filters (not learned one) are sufficient to get very good performances in general in the long range dependencies scenario is a strong one, and as such it needs to have stronger empirical assessment than the one provided in the current version of the paper. This stronger support can only be obtained by using additional and various real-data tasks, and the proposed approach should be compared versus well-established representation learning approaches that learn convolutional filters (or alike).
As a side comment, any theorem that has already been proved in a previous work should be reported with the relevant citation following the Theorem heading, e.g., "Theorem 2.1 [Hazan et al. 2017] statement of the thorem".
A revised version of the paper reporting a stronger empirical results is strongly encouraged. The same reviewers will be asked to review the revised version.

**Audience:**

TMLR's audience will for sure be interested in knowing the findings of this paper, provided the main claim is revised or the experimental assessment is improved.

**Claims And Evidence:**

The paper proposes an interesting and sound approach based on spectral filtering for sequence modeling in the context of  prediction tasks with long range dependencies. The paper is clear and well written. The claim that the spectral filtering approach is effective for tasks requiring very long range memory is not completely supported from the experimental point of view.
All reviewers think that the current empirical evaluation is weak, although two out of three think the present contribution can be worth to be published. Either the authors have to reconsider their claims, or additional experimental evidence, involving real-world data such as genomics/audio tasks as suggested by one of the reviewers, should be reported in order to better support the main claim of the paper. For this reason, I prefer to encourage the authors to resubmit a revision of the paper that addresses this concern.

**Resubmission Of Major Revision:**

The authors may consider submitting a major revision at a later time.